# On the Accuracy of Influence Functions for Measuring Group Effects

**Pang Wei Koh**[*]     **Kai-Siang Ang**[*]     **Hubert H. K. Teo**[*]     **Percy Liang**
Department of Computer Science
Stanford University
{pangwei@cs, kaiang@, hteo@, pliang@cs}.stanford.edu

## Abstract

Influence functions estimate the effect of removing a training point on a model without the need to retrain. They are based on a first-order Taylor approximation that is guaranteed to be accurate for sufficiently small changes to the model, and so are commonly used to study the effect of individual points in large datasets. However, we often want to study the effects of large *groups* of training points, e.g., to diagnose batch effects or apportion credit between different data sources. Removing such large groups can result in significant changes to the model. Are influence functions still accurate in this setting? In this paper, we find that across many different types of groups and for a range of real-world datasets, the predicted effect (using influence functions) of a group correlates surprisingly well with its actual effect, even if the absolute and relative errors are large. Our theoretical analysis shows that such strong correlation arises only under certain settings and need not hold in general, indicating that real-world datasets have particular properties that allow the influence approximation to be accurate.

## 1 Introduction

Influence functions (Jaeckel, 1972; Hampel, 1974; Cook, 1977) estimate the effect of removing an individual training point on a model's predictions without the computationally-prohibitive cost of retraining the model. Tracing a model's output back to its training data can be useful: influence functions have been recently applied to explain predictions (Koh and Liang, 2017), produce confidence intervals (Schulam and Saria, 2019), investigate model bias (Brunet et al., 2018; Wang et al., 2019), improve human trust (Zhou et al., 2019), and even craft data poisoning attacks (Koh et al., 2019).

Influence functions are based on first-order Taylor approximations that are accurate for estimating small perturbations to the model, which makes them suitable for predicting the effects of removing individual training points on the model. However, we often want to study the effects of removing *groups* of points, which represent large perturbations to the data. For example, we might wish to analyze the effect of data collected from different experimental batches (Leek et al., 2010) or demographic groups (Chen et al., 2018); apportion credit between crowdworkers, each of whom generated part of the data (Arrieta-Ibarra et al., 2018); or, in a multi-party learning setting, ensure that no individual user has too much influence on the joint model (Hayes and Ohrimenko, 2018). Are influence functions still accurate when predicting the effects of (removing) these larger groups?

In this paper, we first show empirically that on real datasets and across a broad variety of groups of data, the predicted and actual effects are strikingly *correlated* (Spearman $\rho$ of 0.8 to 1.0), such that the groups with the largest actual effect also tend to have the largest predicted effect. Moreover, the predicted effect tends to *underestimate* the actual effect, suggesting that it could be an approximate

---

[*]Equal contribution.

lower bound in practice. Using influence functions to predict the actual effect of removing large, coherent groups of data can therefore still be useful, even though the violation of the small-perturbation assumption can result in high absolute and relative errors between the predicted and actual effects.

What explains these phenomena of correlation and underestimation? Prior theoretical work focused on establishing the conditions under which this influence approximation is accurate, i.e., the error between the actual and predicted effects is small (Giordano et al., 2019b; Rad and Maleki, 2018). However, in our setting of removing large, coherent groups of data, this error can be quite large. As a first step towards understanding the behavior of the influence approximation in this regime, we characterize the relationship between the predicted and actual effects of a group via the one-step Newton approximation (Pregibon et al., 1981), which we find is a surprisingly accurate approximation in practice. We show that correlation and underestimation arise under certain settings (e.g., removing multiple copies of a single training point), but need not hold in general, which opens up the intriguing question of why we observe those phenomena across a wide range of empirical settings.

Finally, we exploit the correlation of predicted and actual group effects in two example case studies: a chemical-disease relationship (CDR) task, where the groups correspond to different labeling functions (Hancock et al., 2018), and a natural language inference (NLI) task (Williams et al., 2018), where the groups come from different crowdworkers. On the CDR task, we find that the influence of each labeling function correlates with its size (the number of examples it labels) but not its average accuracy, which suggests that practitioners should focus on the coverage of the labeling functions they construct. In contrast, on the NLI task, we find that the influence of each crowdworker is uncorrelated with the number of examples they contibute, which suggests that practitioners should focus on how to elicit high-quality examples from crowdworkers over increasing quantity.

## 2 Background and problem setup

Consider learning a predictive model with parameters $\theta \in \Theta$ that maps from an input space $\mathcal{X}$ to an output space $\mathcal{Y}$. We are given $n$ training points $\{(x_1, y_1), \ldots, (x_n, y_n)\}$ and a loss function $\ell(x, y, \theta)$ that is twice-differentiable and convex in $\theta$. To train the model, we select the model parameters

$$\hat{\theta}(\mathbf{1}) = \arg\min_{\theta \in \Theta} \left[ \sum_{i=1}^n \ell(x_i, y_i; \theta) \right] + \frac{\lambda}{2} \|\theta\|_2^2 \tag{1}$$

that minimize the $L_2$-regularized empirical risk, where $\lambda > 0$ controls regularization strength. The all-ones vector $\mathbf{1}$ in $\hat{\theta}(\mathbf{1})$ denotes that the initial training points all have uniform sample weights.

Our goal is to measure the effects of different groups of training data on the model: if we removed a subset of training points $W$, how much would the model $\hat{\theta}$ change? Concretely, we define a vector $w \in \{0, 1\}^n$ of sample weights with $w_i = \mathbb{I}((x_i, y_i) \in W)$ and consider the modified parameters

$$\hat{\theta}(\mathbf{1} - w) = \arg\min_{\theta \in \Theta} \left[ \sum_{i=1}^n (1 - w_i)\ell(x_i, y_i; \theta) \right] + \frac{\lambda}{2} \|\theta\|_2^2 \tag{2}$$

corresponding to retraining the model after excluding $W$. We refer to $w$ as the subset (corresponding to $W$); the number of removed points as $\|w\|_1$; and the fraction of removed points as $\alpha = \|w\|_1/n$.

The *actual effect* $\mathcal{I}_f^* : [0, 1]^n \to \mathbb{R}$ of the subset $w$ is

$$\mathcal{I}_f^*(w) = f(\hat{\theta}(\mathbf{1} - w)) - f(\hat{\theta}(\mathbf{1})), \tag{3}$$

where the evaluation function $f : \Theta \to \mathbb{R}$ measures a quantity of interest. Specifically, we study:

- The *change in test prediction*, with $f(\theta) = \theta^\top x_{\text{test}}$. Linear models (for regression or binary classification) make predictions that are functions of $\theta^\top x_{\text{test}}$, so this measures the effect that removing a subset will have on the model's prediction for some test point $x_{\text{test}}$.

- The *change in test loss*, with $f(\theta) = \ell(x_{\text{test}}, y_{\text{test}}; \theta)$, which is similar to the test prediction.

- The *change in self-loss*, with $f(\theta) = \sum_{i=1}^n w_i \ell(x_i, y_i; \theta)$, measures the increase in loss on the removed points $w$. Its average over all subsets of size $\|w\|_1$ is the estimated extra loss that leave-$\|w\|_1$-out cross-validation (CV) measures over the training loss.

## 2.1 Influence functions

The issue with computing the actual effect $\mathcal{I}_f^*(w)$ is that retraining the model to compute $\hat{\theta}(\mathbf{1} - w)$ for each subset $w$ can be prohibitively expensive. Influence functions provide a relatively efficient first-order approximation to $\mathcal{I}_f^*(w)$ that avoids retraining.

Consider the function $q_w : [0, 1] \to \mathbb{R}$ with $q_w(t) = f\big(\hat{\theta}(\mathbf{1} - tw)\big)$, such that the actual effect $\mathcal{I}_f^*(w)$ can be written as $q_w(1) - q_w(0)$. We define the *predicted effect* of the subset $w$ to be its *influence* $\mathcal{I}_f(w) = q_w'(0) \approx q_w(1) - q_w(0)$; in this paper, we use the term predicted effect interchangeably with influence. Intuitively, influence measures the effect of removing an infinitesimal weight from each point in $w$ and then linearly extrapolates to removing all of $w$.[2] By taking a Taylor approximation (see, e.g., Hampel et al. (1986) for details), the influence can be computed as

$$\mathcal{I}_f(w) \overset{\text{def}}{=} q_w'(0) = \nabla_\theta f\big(\hat{\theta}(\mathbf{1})\big)^\top \left[ \frac{d}{dt} \hat{\theta}(\mathbf{1} - tw)\Big|_{t=0} \right]$$

$$= \nabla_\theta f\big(\hat{\theta}(\mathbf{1})\big)^\top H_{\lambda,\mathbf{1}}^{-1} g_{\mathbf{1}}(w), \tag{4}$$

where $g_{\mathbf{1}}(w) = \sum_{i=1}^n w_i \nabla_\theta \ell(x_i, y_i; \hat{\theta}(\mathbf{1}))$, $H_{\mathbf{1}} = \sum_{i=1}^n \nabla_\theta^2 \ell(x_i, y_i; \hat{\theta}(\mathbf{1}))$, and $H_{\lambda,\mathbf{1}} = H_{\mathbf{1}} + \lambda I$. When measuring the change in test prediction or test loss, influence is additive: if $w = w_1 + w_2$, then $\mathcal{I}_f(w) = \mathcal{I}_f(w_1) + \mathcal{I}_f(w_2)$, i.e., the influence of a subset is the sum of influences of its constituent points, and we can efficiently compute the influence of any subset by pre-computing the influence of each individual point (e.g., by taking a single inverse Hessian-vector product, as in Koh and Liang (2017)). However, when measuring the change in self-loss, influence is not additive and requires a separate calculation for each subset removed.

## 2.2 Relation to prior work

Influence functions—introduced in the seminal work of Hampel (1974) and in Jaeckel (1972), where it was called the infinitesimal jackknife—have a rich history in robust statistics. The use of influence functions in the ML community is more recent, though growing; in Section 1, we provide references for several recent applications of influence functions in ML.

Removing a single training point, especially when the total number of points $n$ is large, represents a small perturbation to the training distribution, so we expect the first-order influence approximation to be accurate. Indeed, prior work on the accuracy of influence has focused on this regime: e.g., Debruyne et al. (2008); Liu et al. (2014); Rad and Maleki (2018); Giordano et al. (2019b) give evidence that the influence on self-loss can approximate LOOCV, and Koh and Liang (2017) similarly examined the accuracy of estimating the change in test loss after removing single training points.

However, removing a constant fraction $\alpha$ of the training data represents a large perturbation to the training distribution. To the best of our knowledge, this setting has not been empirically studied; perhaps the closest work is Khanna et al. (2019)'s use of Bayesian quadrature to estimate a maximally influential subset. Instead, older references have alluded to the phenomena of correlation and underestimation we observe: Pregibon et al. (1981) note that influence tends to be conservative, while Hampel et al. (1986) say that "bold extrapolations" (i.e., large perturbations) are often still useful. On the theoretical front, Giordano et al. (2019b) established finite-sample error bounds that apply to groups, e.g., showing that the leave-$k$-out approximation is consistent as the fraction of removed points $\alpha \to 0$. Our focus is instead on the relationship of the actual effect $\mathcal{I}_f^*(w)$ and predicted effect (influence) $\mathcal{I}_f(w)$ in the regime where $\alpha$ is constant and the error $|\mathcal{I}_f^*(w) - \mathcal{I}_f(w)|$ is large.

## 3 Empirical accuracy of influence functions on constructed groups

How well do influence functions estimate the effect of (removing) a group of training points? If $n$ is large and we remove a subset $w$ uniformly at random, the new parameters $\hat{\theta}(\mathbf{1} - w)$ should remain close to $\hat{\theta}(\mathbf{1})$ even when if fraction of removed points $\alpha$ is non-negligible, so the influence error $|\mathcal{I}_f^*(w) - \mathcal{I}_f(w)|$ should be small. However, we are usually interested in removing coherent, non-random groups, e.g., all points from a data source or share some feature. In such settings, the

| Dataset | Classes | $n$ | $d$ | $\lambda/n$ | Test acc. | Source |
|---------|---------|-----|-----|-------------|-----------|--------|
| Diabetes | 2 | 20,000 | 127 | $2.2 \times 10^{-4}$ | 68.2% | Strack et al. (2014) |
| Enron | 2 | 4,137 | 3,289 | $1.0 \times 10^{-3}$ | 96.1% | Metsis et al. (2006) |
| Dogfish | 2 | 1,800 | 2,048 | $2.2 \times 10^{-2}$ | 98.5% | Koh and Liang (2017) |
| MNIST | 10 | 55,000 | 784 | $1.0 \times 10^{-3}$ | 92.1% | LeCun et al. (1998) |
| CDR | 2 | 24,177 | 328 | $1.0 \times 10^{-4}$ | 67.4% | Hancock et al. (2018) |
| MultiNLI | 3 | 392,702 | 600 | $1.0 \times 10^{-4}$ | 50.4% | Williams et al. (2018) |

Table 1: Dataset characteristics and the test accuracies that logistic regression achieves (with regularization $\lambda$ selected by cross-validation). $n$ is the training set size and $d$ is the number of features.

parameters $\hat{\theta}(\mathbf{1} - w)$ and $\hat{\theta}(\mathbf{1})$ might differ substantially, and the error $|\mathcal{I}_f^*(w) - \mathcal{I}_f(w)|$ could be large. Put another way, there could be a cluster of points such that removing one of those points would not change the model by much—so influence could be low—but removing all of them would.

Surprisingly (to us), we found that even when removing large and coherent groups of points, the influence $\mathcal{I}_f(w)$ behaved consistently relative to the actual effect $\mathcal{I}_f^*(w)$ on test predictions, test losses, and self-loss, with two broad phenomena emerging:

1. **Correlation**: $\mathcal{I}_f(w)$ and $\mathcal{I}_f^*(w)$ rank subsets of points $w$ similarly (e.g., high Spearman $\rho$).

2. **Underestimation**: $\mathcal{I}_f(w)$ and $\mathcal{I}_f^*(w)$ tend to have the same sign with $|\mathcal{I}_f(w)| < |\mathcal{I}_f^*(w)|$.[3]

Here, we report results on 5 datasets chosen to span a range of applications, training set size $n$, and number of features $d$ (Table 1).[4] In an attempt to make the influence approximation as inaccurate as possible, we constructed a variety of subsets, from small ($\alpha = 0.25\%$) to large ($\alpha = 25\%$), to be coherent and have considerable influence on the model. On each dataset, we trained an $L_2$-regularized logistic regression model (or softmax for the multiclass tasks) and compared the influences and actual effects of these subsets.

**Group construction.** Our aim is to construct coherent groups that when removed will substantially change the model. To do so, we need to choose points that are similar in some way. Specifically, for each dataset, we grouped points in 7 ways: 1) points that share feature values; 2) points that cluster on their features or 3) on their gradients $\nabla_\theta \ell(x, y, \hat{\theta}(\mathbf{1}))$; 4) random points within the same class; 5) random points from any class. We also grouped 6) points with large positive and 7) negative influence on the test loss $\ell(x_{\text{test}}, y_{\text{test}}, \hat{\theta}(\mathbf{1}))$, since intuitively, training points that all have high influence on a test point should act together to change the model substantially. Overall, for each dataset, we constructed 1,700 subsets ranging in size from $0.25\%$ to $25\%$ of the training points. See Appendix A for more details.

**Results.** Figure 1 shows that the influences and actual effects of all of these subsets on test prediction (Top), test loss (Mid), and self-loss (Bot) are highly correlated (Spearman $\rho$ of 0.89 to 0.99 across all plots), even though the absolute and relative errors of the influence approximation can be quite large. Moreover, the influence of a group tends to underestimate its actual effect in all settings except for groups with negative influence on test loss (the left side of each plot in Figure 1-Mid). These trends held across a wide range of regularizations $\lambda$, though correlation increased with $\lambda$ (Appendix C.2).

In Section 5, we will use the CDR dataset (Hancock et al., 2018) and the MultiNLI (Williams et al., 2018) dataset to show that correlation and underestimation also apply to groups of data that arise naturally, and that influence functions can therefore be used to derive insights about real datasets and applications. Before that, we first attempt to develop some theoretical insight into the results above.

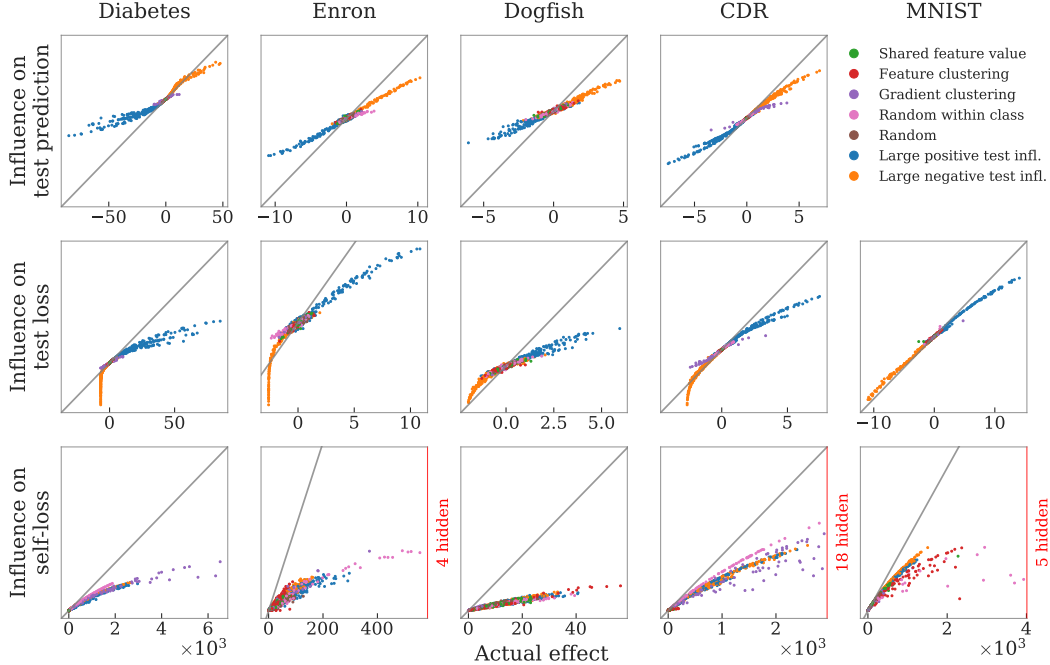

Figure 1: Influences vs. actual effects of coherent groups of points ranging from $0.25\%$ to $25\%$ in size. Each point corresponds to a group, and its color reflects how that group was constructed. In Top and Mid, we show results for the test point with highest loss; other test points are similar (Appendix C.1), though with more curvature for test loss (Appendix C.3). The grey reference line has slope 1, and the red borders represent points that are not plotted because they are outside the x- or y-axis range. We omit the top row for MNIST, as $\theta^\top x_{\text{test}}$ is not meaningful in the multi-class setting.

## 4 Theoretical analysis

The experimental results above show that there is consistent underestimation and high correlation between the predicted effects, based on influence functions, and the actual effects of groups across a variety of datasets, despite the influence approximation incurring large absolute and relative error. As we discussed in Section 2.2, this is outside the regime of existing theory.

As an initial step towards understanding the high-error regime, we establish conditions under which the actual effect $\mathcal{I}_f^*(w)$ lies approximately between $\mathcal{I}_f(w)$ and $C_{\max}\mathcal{I}_f(w)$ for some $C_{\max} > 0$. This cone constraint—so called because it implies that all points on the graph of influence vs. actual effect lie within a cone—implies underestimation and, if $C_{\max}$ is small, some degree of correlation. We first show that this constraint holds in restricted settings—when measuring self-loss, or when removing multiple copies of the same point—and that $C_{\max}$ varies inversely with the regularization term $\lambda$, which is expected since stronger regularization reduces the change in the model. However, the cone constraint is stronger than necessary because it bounds the degree of underestimation, and we construct counterexamples to show that it need not hold in more general settings.

Our analysis centers on the *one-step Newton approximation*, which estimates the change in parameters

$$\hat{\theta}(\mathbf{1} - w) - \hat{\theta}(\mathbf{1}) \approx \Delta\theta_{\text{Nt}}(w) \stackrel{\text{def}}{=} \left(H_{\lambda,\mathbf{1}}(\mathbf{1} - w)\right)^{-1} g_{\mathbf{1}}(w),$$

where $H_{\lambda,\mathbf{1}}(\mathbf{1} - w) = \left(\sum_{i=1}^{n}(1 - w_i)\nabla_\theta^2 \ell(x_i, y_i; \hat{\theta}(\mathbf{1}))\right) + \lambda I$ is the regularized empirical Hessian at $\hat{\theta}(\mathbf{1})$ but reweighted after removing the subset $w$. This change in parameters gives the Newton approximation of the effect $\mathcal{I}_f^{\text{Nt}}(w) = f\left(\hat{\theta}(\mathbf{1}) + \Delta\theta_{\text{Nt}}(w)\right) - f(\hat{\theta}(\mathbf{1})))$ and the corresponding Newton error $\text{Err}_{\text{Nt-act}}(w) = \mathcal{I}_f^*(w) - \mathcal{I}_f^{\text{Nt}}(w)$, which measures its gap from the actual effect. Specifically, we decompose the error between the actual effect $\mathcal{I}_f^*(w)$ and influence $\mathcal{I}_f(w)$ as

$$\mathcal{I}_f^*(w) - \mathcal{I}_f(w) = \underbrace{\mathcal{I}_f^*(w) - \mathcal{I}_f^{\text{Nt}}(w)}_{\text{Err}_{\text{Nt-act}}(w)} + \underbrace{\mathcal{I}_f^{\text{Nt}}(w) - \mathcal{I}_f(w)}_{\text{Err}_{\text{Nt-inf}}(w)}. \tag{5}$$

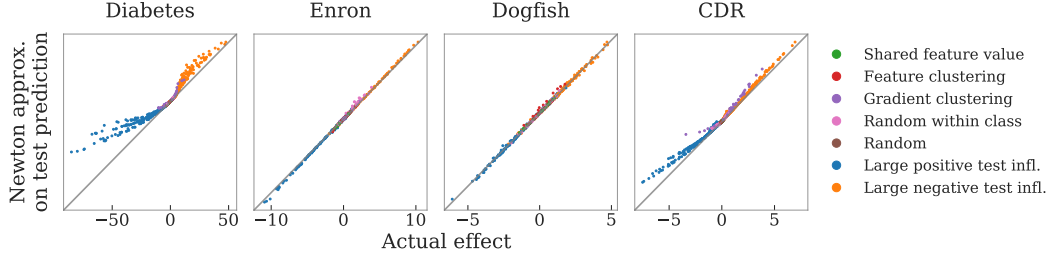

Figure 2: The Newton approximation accurately captures the actual effect for our datasets (though there is more error on the Diabetes dataset), with the same test point as in Figure 1-Top. We omit MNIST and MultiNLI for computational reasons. See Figure C.4 for plots of test loss and self-loss.

In Section 4.1, we first show that the Newton-actual error $\text{Err}_{\text{Nt-act}}(w)$ decays at a rate of $O\big(1/(\sigma_{\min} + \lambda)^3\big)$, where $\lambda$ is regularization strength and $\sigma_{\min}$ is the smallest eigenvalue of the empirical Hessian $H_{\mathbf{1}}$. Empirically, this error is small on our datasets, so we focus on characterizing the Newton-influence error $\text{Err}_{\text{Nt-inf}}(w)$ in Section 4.2. We use this characterization to study the behavior of influence relative to the actual effect on self-loss (Section 4.3) and test prediction (Section 4.4). For margin-based models, the test loss is a monotone function of the test prediction, so the analysis is similar (Appendix D.3).

## 4.1 Bounding the error of the one-step Newton approximation

The Newton approximation is computationally expensive because it computes $(H_{\lambda, \mathbf{1}}(\mathbf{1} - w))^{-1}$ for each $w$ (instead of the fixed $H_{\lambda, \mathbf{1}}^{-1}$ in the influence calculation). However, it provides more accurate estimates (e.g., Pregibon et al. (1981), Rad and Maleki (2018)), and we show that its error can be bounded as follows (all proofs in Appendix E):

**Proposition 1.** *Let the Newton error be* $\text{Err}_{\text{Nt-act}}(w) \overset{\text{def}}{=} \mathcal{I}_f^*(w) - \mathcal{I}_f^{\text{Nt}}(w)$. *Assume that the evaluation function* $f(\theta)$ *is* $C_f$-*Lipschitz and that the Hessian* $\nabla_\theta^2 \ell(x, y, \theta)$ *is* $C_H$-*Lipschitz. Then*

$$|\text{Err}_{\text{Nt-act}}(w)| \leq \frac{n\|w\|_1^2 C_f C_H C_\ell^2}{(\sigma_{\min} + \lambda)^3},$$

*where we define* $C_\ell \overset{\text{def}}{=} \max_{1 \leq i \leq n} \|\nabla_\theta \ell(x_i, y_i, \hat{\theta}(\mathbf{1}))\|_2$ *to be the largest norm of a training point's gradient at* $\hat{\theta}(\mathbf{1})$, *and* $\sigma_{\min}$ *to be the smallest eigenvalue of* $H_{\mathbf{1}}$. $\text{Err}_{\text{Nt-act}}(w)$ *only involves third-order or higher derivatives of the loss, so it is 0 for quadratic losses.*

Proposition 1 tells us that the Newton approximation is accurate when $\lambda$ is large or the third derivative of $\ell(x, y; \cdot)$ (controlled by $C_H$) is small. Empirically, the Newton error $\text{Err}_{\text{Nt-act}}(w)$ is strikingly small in most of our settings (Figure 2), even though the overall error of the influence approximation $\mathcal{I}_f^*(w) - \mathcal{I}_f(w)$ is still large. In the remainder of this section, we therefore focus on characterizing the Newton-influence error $\text{Err}_{\text{Nt-inf}}(w)$, under the assumption that the Newton approximation is similar to the actual effect (within a factor of $O(1/\lambda^3)$).

## 4.2 Characterizing the difference between the Newton approximation and influence

We next characterize the Newton-influence error $\text{Err}_{\text{Nt-inf}}(w) = \mathcal{I}_f^{\text{Nt}}(w) - \mathcal{I}_f(w)$:

**Proposition 2.** *Under the assumptions of Proposition 1 and the additional assumption that the third derivative of* $f(\theta)$ *exists and is bounded in norm by* $C_{f,3}$, *the Newton-influence error* $\text{Err}_{\text{Nt-inf}}(w)$ *is*

$$\text{Err}_{\text{Nt-inf}}(w) = \nabla_\theta f(\hat{\theta}(\mathbf{1}))^\top H_{\lambda, \mathbf{1}}^{-\frac{1}{2}} D(w) H_{\lambda, \mathbf{1}}^{-\frac{1}{2}} g_{\mathbf{1}}(w) + \underbrace{\frac{1}{2}\Delta\theta_{\text{Nt}}(w)^\top \nabla_\theta^2 f(\hat{\theta}(\mathbf{1}))\Delta\theta_{\text{Nt}}(w) + \text{Err}_{f,3}(w)}_{\textit{Error from curvature of } f(\cdot)},$$

*with* $D(w) \overset{\text{def}}{=} \big(I - H_{\lambda, \mathbf{1}}^{-\frac{1}{2}} H_{\mathbf{1}}(w) H_{\lambda, \mathbf{1}}^{-\frac{1}{2}}\big)^{-1} - I$ *and* $H_{\mathbf{1}}(w) \overset{\text{def}}{=} \sum_{i=1}^n w_i \nabla_\theta^2 \ell(x_i, y_i; \hat{\theta}(\mathbf{1}))$. *The error matrix* $D(w)$ *has eigenvalues between 0 and* $\frac{\sigma_{\max}}{\lambda}$, *where* $\sigma_{\max}$ *is the largest eigenvalue of* $H_{\mathbf{1}}$. *The*

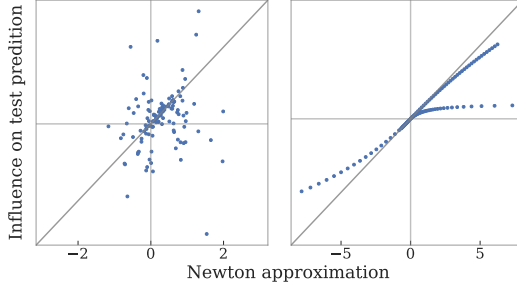

**Figure 3:** Influence $\mathcal{I}_f(w)$ vs. Newton approximation $\mathcal{I}_f^{\mathrm{Nt}}(w)$ on the test prediction on two counterexamples detailed in Appendix D.1. Left: We adversarially choose a set of $w$'s such that $\mathcal{I}_f(w)$ and $\mathcal{I}_f^{\mathrm{Nt}}(w)$ can have different signs and need not correlate. Right: When we only remove copies of single points, underestimation holds. However, we can control the scaling factor $d(w)$ between $\mathcal{I}_f(w)$ and $\mathcal{I}_f^{\mathrm{Nt}}(w)$ on different groups, so correlation need not hold.

*residual term* $\mathrm{Err}_{f,3}(w)$ *captures the error due to third-order derivatives of* $f(\cdot)$ *and is bounded by* $|\mathrm{Err}_{f,3}(w)| \le \|w\|_1^3 C_{f,3} C_\ell^3 / 6(\sigma_{\min} + \lambda)^3$.

We can interpret Proposition 2 as a formalization of Hampel et al. (1986)'s observation that influence approximations are accurate when the model is robust and the curvature of the loss is low. In general, the error decreases as $\lambda$ increases and $f(\cdot)$ becomes less curved; in Figure C.2, we show that increasing $\lambda$ reduces error and increases correlation in our experiments.

### 4.3 The relationship between influence and actual effect on self-loss

Let us now apply Proposition 2 to analyze the behavior of influence under different choices of evaluation function $f(\cdot)$. We start with the self-loss $f(\theta) = \sum_{i=1}^n w_i \ell(x_i, y_i; \theta)$, as its influences and actual effects are always non-negative, and it is the cleanest to characterize:

**Proposition 3.** *Under the assumptions of Proposition 2, the influence on the self-loss obeys*

$$\mathcal{I}_f(w) + \mathrm{Err}_{f,3}(w) \le \mathcal{I}_f^{\mathrm{Nt}}(w) \le \left(1 + \frac{3\sigma_{\max}}{2\lambda} + \frac{\sigma_{\max}^2}{2\lambda^2}\right) \mathcal{I}_f(w) + \mathrm{Err}_{f,3}(w).$$

The constraint in Proposition 3 implies that up to $O(1/\lambda^3)$ terms, influence underestimates the Newton approximation and therefore the actual effect. This explains the previously-unexplained downward bias observed when using influence to approximate LOOCV (Debruyne et al., 2008; Giordano et al., 2019b). Equivalently, all points on the graph of influences vs. actual effects lie within the cone bounded by the lines with slope 1 and slope $\frac{\lambda}{\lambda + 3\sigma_{\max}/2}$ lines, up to $O(1/\lambda^3)$ terms. As $\lambda$ grows, these lines will converge, and the error terms $\mathrm{Err}_{f,3}(w)$ and $\mathrm{Err}_{\mathrm{Nt-act}}(w)$ will decay at a rate of $O(1/\lambda^3)$, forcing the influences and actual effects to be equal.

However, $\lambda/\sigma_{\max}$ is quite small in our experiments in Section 3, so the actual correlation of influence is better than predicted by this theory: in Figure 1-Bot, the sizes of the theoretically-permissible cones can be quite large, but the points in the graphs nevertheless trace a tight curve through the cone.

### 4.4 The relationship between influence and actual effect on a test point

We now turn to measuring the test prediction $f(\theta) = \theta^\top x_{\mathrm{test}}$. Here, we show that correlation and underestimation need not hold, and that we cannot obtain a cone constraint similar to Proposition 3 except in a restricted setting. Define $v_{\mathrm{test}} = H_{\lambda,\mathbf{1}}^{-\frac{1}{2}} x_{\mathrm{test}}$ and $v_w = H_{\lambda,\mathbf{1}}^{-\frac{1}{2}} g_{\mathbf{1}}(w)$. Proposition 2 gives:

**Corollary 1.** *Suppose* $f(\theta) = \theta^\top x_{\mathrm{test}}$. *Then* $\mathcal{I}_f^{\mathrm{Nt}}(w) = \mathcal{I}_f(w) + v_{\mathrm{test}}^\top D(w) v_w$, *where* $D(w) = \left(I - H_{\lambda,\mathbf{1}}^{-\frac{1}{2}} H_{\mathbf{1}}(w) H_{\lambda,\mathbf{1}}^{-\frac{1}{2}}\right)^{-1} - I$ *is the* error matrix *from Proposition 2.*

Unfortunately, Corollary 1 implies that no cone constraint applies: in general, we can find $x_{\mathrm{test}}$ such that the influence $\mathcal{I}_f(w) = v_{\mathrm{test}}^\top v_w = 0$ but the Newton approximation $\mathcal{I}_f^{\mathrm{Nt}}(w) = v_{\mathrm{test}}^\top D(w) v_w$ is large. As a counterexample, Figure 3-Left shows that on synthetic data, $\mathcal{I}_f(w)$ and $\mathcal{I}_f^{\mathrm{Nt}}(w)$ can even have opposite signs on some subsets $w$.

We can recover a cone constraint similar to Proposition 3 if we restrict our attention to the special case where we use a margin-based model and remove (possibly multiple copies) of a single point:

**Proposition 4.** *Consider a binary classification setting with $y \in \{-1, +1\}$ and a margin-based model with loss $\ell(x, y; \theta) = \phi(y\theta^\top x)$ for some $\phi : \mathbb{R} \to \mathbb{R}_+$. Suppose $f(\theta) = \theta^\top x_{\text{test}}$ and that the subset $w$ comprises $\|w\|_1$ identical copies of the training point $(x_w, y_w)$. Then under the assumptions of Proposition 1, the Newton approximation $\mathcal{I}_f^{\text{Nt}}(w)$ is related to the influence $\mathcal{I}_f(w)$ according to*

$$\mathcal{I}_f^{\text{Nt}}(w) = \frac{\mathcal{I}_f(w)}{1 - \|w\|_1 \cdot \phi''(y_w\hat{\theta}(\mathbf{1})^\top x_w) \cdot x_w^\top H_{\lambda,\mathbf{1}}^{-1} x_w}.$$

*This implies the Newton approximation $\mathcal{I}_f^{\text{Nt}}(w)$ is bounded between $\mathcal{I}_f(w)$ and $\left(1 + \frac{\sigma_{\max}}{\lambda}\right)\mathcal{I}_f(w)$.*

Similar to Proposition 3, Proposition 4 shows that up to $O(1/\lambda^3)$ terms, the influence underestimates the actual effect when removing copies of a single point. Moreover, all points on the graph of influences vs. actual effects lie within the cone bounded by the lines with slope 1 and slope $\lambda/(\lambda + \sigma_{\max})$, up to $O(1/\lambda^3)$ terms. As $\lambda/\sigma_{\max}$ grows, the cone shrinks, and correlation increases.

However, if $\lambda/\sigma_{\max}$ is small (as in our experiments in Section 3), the cone is wide, and the scaling factor $d(w) = 1/(1 - \|w\|_1 \cdot \phi''_k x_k^\top H_{\lambda,\mathbf{1}}^{-1} x_k)$ in Proposition 4 can be quite large for some subsets $w$ but not for others. In particular, $d(w)$ is large when there are few remaining points in the direction of the removed points. In Figure 3-Right, we exploit this fact to show that the influence $\mathcal{I}_f(w)$ and Newton approximation $\mathcal{I}_f^{\text{Nt}}(w)$ can exhibit low correlation (e.g., low $\mathcal{I}_f(w)$ need not mean low $\mathcal{I}_f^{\text{Nt}}(w)$), even in the simplified setting of removing copies of single points. We comment on the analogue of $d(w)$ in the general multiple-point setting in Appendix D.2, and on the influence on test loss (instead of test prediction) in Appendix D.3.

## 5 Applications of influence functions on natural groups of data

The analysis in Section 4 shows that the cone constraint between predicted and actual group effects need not always hold. Nonetheless, our experiments in Section 3 demonstrate that on real datasets, the correlation is much stronger than the theory predicts. We now turn to using influence functions to predict group effects in two case studies where groups arise naturally.

**Chemical-disease relation (CDR).** The CDR dataset tackles the following task: given text about the relationship between a chemical and a disease, predict if the chemical causes the disease. It was collected via data programming, where users provide labeling functions (LFs)—instead of labels—that take in an unlabeled point and either abstain or output a heuristic label (Ratner et al., 2016). Specifically, Hancock et al. (2018) collected natural language explanations of provided classifications; parsed those explanations into LFs; and used those LFs to label a large pool of data (Appendix B.1).

We used influence functions to study two important properties of LFs: *coverage*, the fraction of unlabeled points for which an LF outputs a non-abstaining label; and *precision*, the proportion of correct labels output. We associated each LF with the group of points that it labeled, and computed its influence; as expected, these correlated with actual effects on overall test loss (Spearman $\rho = 1$; Figure C.5). LFs with higher coverage had more influence (Figure 4-Left; see also Figure C.6), but surprisingly, LFs with higher precision did not (Figure 4-Mid). The association with coverage stems at least partially from class balance: each LF outputs either all positive or all negative labels, so removing an LF with high coverage changes the class balance and consequently improves test performance on one class at the expense of the other (Figure 4-Left). While these findings are not causal claims, they suggest that the coverage of an LF, rather than its precision, might have a stronger effect on its overall contribution to test performance.

**MultiNLI.** The MultiNLI dataset deals with natural language inference: determining if a pair of sentences agree, contradict, or are neutral. Williams et al. (2018) presented crowdworkers with initial sentences from five genres and asked them to generate follow-on sentences that were neutral or in agreement/contradiction (Appendix B.2). We studied the effect that each crowdworker had on the model's test set performance by computing the influence of the examples they created on overall test loss (Spearman $\rho$ of 0.77 to 0.86 with actual effects across different genres; see Figure C.8).

Studying the influence of each crowdworker reveals that the number of examples a crowdworker created was not predictive of influence on test performance: e.g., the most prolific crowdworker

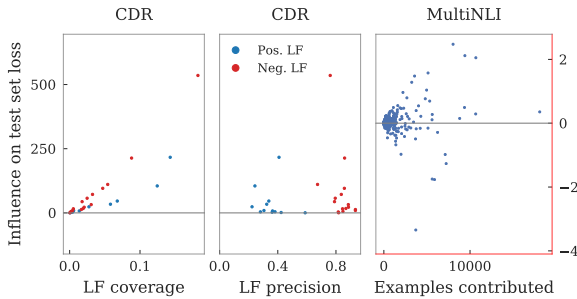

Figure 4: In CDR, the influence of a labeling function (LF) on test performance is predicted by its coverage (Left) but not its precision (Mid). However, in MultiNLI, the number of examples contributed by a crowdworkers is not predictive of its influence (Right). For CDR, LFs output either all + or all − labels; we plot the influence of each LF on the test points of the same class.

contributed 35,000 examples but had negative influence, and we verified that removing all of those examples and retraining the model indeed made overall test performance worse (Figure 4-Right). Curiously, this effect was genre-specific: crowdworkers who improved performance on some genres would lower performance on others (Figure C.10), even though the number of examples they contributed to a genre did not correlate with their influence on it (Figure C.11). We note that these results are obtained on a baseline logistic regression model built on top of a continuous bag-of-words representation. Identifying precisely what makes a crowdworker's contributions useful, especially on higher-performing models, could help us improve dataset collection and credit attribution as well as better understand the biases due to annotator effects (Geva et al., 2019).

## 6 Discussion

In this paper, we showed empirically that the influences of groups of points are highly correlated with, and consistently underestimate, their actual effects across a range of datasets, types of groups, and sizes. These phenomena allows us to use influence functions to better understand the "different stories that different parts of the data tell," in the words of Hampel et al. (1986). We showed that we can gain insight into the effects of a labeling function in data programming, or a crowdworker in a crowdsourced dataset, by computing the influence of their corresponding group effects.

While these applications involved predefined groups, influence functions could potentially also discover coherent, semantically-relevant groups in the data. They can also be used to approximate Shapley values, which are a different but related way of measuring the effect of data points; see, e.g., Jia et al. (2019) and Ghorbani and Zou (2019). Separately, influence functions can also estimate the effects of *adding* training points. In this context, underestimation turns into overestimation, i.e., the influence of adding a group of training points tends to overestimate the actual effect of adding that group. This raises the possibility of using influence functions to evaluate the vulnerability of a given dataset and model to data poisoning attacks (Steinhardt et al., 2017).

Our theoretical analysis showed that while correlation and underestimation hold in some restricted settings, they need not hold in general, realistic settings. This gap between theory and experiments opens up important directions for future work: Why do we observe such striking correlation between predicted and actual effects on real data? To what extent is this due to the specific model, datasets, or subsets used? Do these trends hold for non-convex models like neural networks? Our work suggests that there could be distributional assumptions that hold for real data and give rise to the broad phenomena of correlation and underestimation. One promising lead is the surprising observation that the Newton approximation is much more accurate than influence at predicting group effects, which holds out the hope that we can understand group effects using just low-order terms (since the Newton approximation only uses the first and second derivatives of the loss) without needing to account for the whole loss function through higher order terms (as in Giordano et al. (2019a)).

**Reproducibility**

The code for replicating our experiments is available in the GitHub repository `https://github.com/kohpangwei/group-influence-release`. An executable version of this paper is also available on CodaLab at `https://worksheets.codalab.org/worksheets/0xfed2ae0b9e5b44b7a1af8096365592a5`.

**Acknowledgments**

We are grateful to Zhenghao Chen, Brad Efron, Jean Feng, Tatsunori Hashimoto, Robin Jia, Stephen Mussmann, Aditi Raghunathan, Marco Túlio Ribeiro, Noah Simon, Jacob Steinhardt, and Jian Zhang for helpful discussions and comments. We are further indebted to Ryan Giordano, Ruoxi Jia, and Will Stephenson for discussion about prior work, and Samuel Bowman, Braden Hancock, Emma Pierson, and Pranav Rajpurkar for their assistance with applications and datasets. This work was funded by an Open Philanthropy Project Award. PWK was supported by the Facebook Fellowship Program.

## Footnotes

[2]In the statistics literature, influence typically refers to the effect of *adding* weight, so the sign is flipped.

[3] This holds with one exception: when measuring the change in test loss, $f(\theta) = \ell(x_{\text{test}}, y_{\text{test}}; \theta)$, underestimation only holds when actual effect $\mathcal{I}_f^*(w)$ is positive (Figure 1-Mid).

[4] The first 4 datasets involve hospital readmission prediction, spam classification, and object recognition, and were used in Koh and Liang (2017) to study the influence of individual points. The fifth dataset is a chemical-disease relationship (CDR) dataset Hancock et al. (2018). In Section 5, we will also study the MultiNLI language inference dataset (Williams et al., 2018), which was omitted from the experiments here because its large size makes repeated retraining to compute the actual effect too expensive. See Appendix B for dataset details.

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
