[Supplementary Material · supplement.pdf]

# A  Experimental details for comparing influence vs. actual effects on constructed groups

## A.1  Model training

For all experiments in Section 3, we trained a logistic regression model (or softmax for multiclass) using `sklearn.linear_model.LogisticRegression.fit`, fitting the intercept but only applying $L_2$-regularization to the weights. To choose the regularization strength $\lambda$, we conducted 5-fold cross-validation across 10 possible values of $\lambda/n$ logarithmically spaced between $1.0 \times 10^{-4}$ and $1.0 \times 10^{-1}$, inclusive, selecting the regularization that yielded the highest cross-validation accuracy (except on the CDR dataset, where we selected regularization based on cross-validation F1 score to account for class imbalance as per Hancock et al. (2018)'s procedure).

## A.2  Group construction

For each dataset, we constructed groups of various sizes relative to the entire dataset by considering 100 sizes linearly spaced between $0.25\%$ and $25\%$ of the dataset. For each of these 100 sizes, we constructed one group with each of the following methods:

1. Shared features: We selected a single feature uniformly at random and sorted the dataset along this selected feature. Next, we selected an training point uniformly at random. We then constructed a group of size $s$ that consisted of the $s$ unique training points that were closest to the chosen point, as measured by their values in the selected feature. We randomly sampled a feature and initial training point for each different group constructed in this way.

2. Feature clustering: We clustered the dataset with respect to raw features via `scipy.cluster.hierarchy.fclusterdata` with `t` set to 1, as well as with `sklearn.cluster.KMeans.fit` with `n_clusters` taking on values $4, 8, 16, 32, 64, 128$. Since hierarchical clustering determines cluster sizes automatically with a principled heuristic and we try a range of values for `n_clusters` in $k$-means, this recovers clusters with a large range of sizes. The clustering with `n_clusters` $= 4$ also guarantees (via the pigeonhole principle) that there is at least one cluster which contains at least $25\%$ of the dataset. From all the clusters that are at least the size of the desired group, we chose one uniformly at random and chose the group uniformly at random and without replacement from the training points in this cluster.

3. Gradient clustering: We followed the same procedure as "Feature clustering," except that we clustered the dataset with respect to $\nabla_\theta \ell(x, y; \hat{\theta}(\mathbf{1}))$, i.e. each training point was represented by the gradient of the loss on that point.

4. Random within class: We considered all classes with at least as many training points as the size of the desired group. From these classes, we chose one uniformly at random. Then, we chose the group uniformly at random and without replacement from all training points in this class.

5. Random: We picked a group uniformly at random and without replacement from the entire dataset.

The above methods gave us a total of 500 groups (100 groups per method) for each dataset, with the exception of the "random within class" method for MNIST. Since MNIST has 10 classes, each with only $10\%$ of the data, we skipped over groups of size $> 10\%$ just for the "random within class" groups.

In addition, we selected 3 random test points and the 3 test points with highest loss; we intend these to represent the average case and the more extreme case that may be relevant to model developers who want to debug errors that their model outputs. For each of these 6 test points, we selected groups that had large positive influence on its test loss. More specifically, we proceeded in 3 stages:

1. We considered 33 group sizes linearly spaced between $0.25\%$ and $2.5\%$ of the dataset, and for any size $s$ out of these 33, we selected a group uniformly at random and without replacement from training points in the top $1.5 \times 2.5\%$ of the dataset, ordered according to their influence on the test point of interest.

2. This was similar to the first stage, but with 33 sizes spaced between $0.25\%$ and $10\%$ and groups chosen from the top $1.5 \times 10\%$ of the dataset.

3. Finally, we considered 34 sizes spaced between $0.25\%$ and $25\%$, with groups chosen from the top $1.5 \times 25\%$ of the dataset.

Larger groups tend to have lower average influence than smaller groups, since by necessity, the group must contain points farther from the top. This multi-stage approach ensured that we would select small groups with both a high average influence and also with a low average influence, so that we could compare them to larger groups and mitigate confounding the group size with its average influence.

Finally, we repeated this last method of group construction for groups with large negative influence on test point loss.

Using these 6 test points, we generated 1,200 groups (100 subsets per group, with 6 test points, and drawing from the positive and negative tails). In total, we therefore generated 1,700 groups per dataset (except MNIST).

### A.3 Comparison of influence and actual effect

To produce Figure 1, we selected groups as described in Appendix A.2. We retrained the model once for each group, excluding the group in order to calculate its actual effect. To compute all groups' influences, we first calculated the influence of every individual training point using the procedure of Koh and Liang (2017). Then, to compute the influence on test prediction or loss of some group, we simply added the relevant individual influences (in CDR, we weighted these individual influences according to that point's weight; see Appendix B.1). To compute the influence on self-loss of some group, we summed up the gradients of the loss of each training point to compute $g_{\mathbf{1}}(w)$, we calculated the inverse Hessian vector product $H_{\lambda,\mathbf{1}}^{-1}g_{\mathbf{1}}(w)$ and took its dot product with $g_{\mathbf{1}}(w)$ (again, we modified this with appropriate weighting for individual points in CDR).

## B Dataset details

We used the same versions of the Diabetes, Enron, Dogfish, and MNIST datasets as Koh and Liang (2017), since the examination of the accuracy of influence functions for large perturbations is a natural extension of their studies of small perturbations. Additionally, we applied influence to more natural settings in CDR and MultiNLI; here, we discuss their preprocessing pipelines.

### B.1 CDR

Hancock et al. (2018) established the BabbleLabble framework for data programming, following the following pipeline: They took labeled examples with natural language explanations, parsed the explanations into programmatic labeling functions (LFs) via a semantic parser, and filtered out obviously incorrect LFs. Then, they applied the remaining LFs to unlabeled data to create a sparse label matrix, from which they learned a label aggregator that outputs a noisily labeled training set. Finally, they ran $L_2$-regularized logistic regression on a set of basic linguistic features with the noisy labels.

They demonstrated their method on three datasets: Spouse, CDR, and Protein. The Protein dataset was not publicly available, and the vast majority of Spouse was labeled by a single LF, hence we chose to use CDR. This dataset's associated task involved identifying whether, according to a given sentence, a given chemical causes a given disease. For instance, the sentence "Young women on replacement *estrogens* for ovarian failure after cancer therapy may also have increased risk of *endometrial carcinoma* and should be examined periodically." would be labeled True, since it indicates that estrogens may cause endometrial carcinoma (Hancock et al., 2018). The sentences and ground truth labels were sourced from the 2015 BioCreative chemical-disease relation dataset (Wei et al., 2015).

In our application, we began with their 28 LFs and the corresponding label matrix. For simplicity, we did not learn a label aggregator; instead, if an example $x$ was given labels $y_{i_1}, y_{i_2}, \ldots, y_{i_k}$ by $k$ LFs $i_1, \ldots, i_k$, then we created $k$ copies of $x$, each with weight $1/k$. The subset of points corresponding

to LF $i_1$ then included one instance of $x$ with weight $1/k$. This weighting was taken into account in model training as well as in calculations of influence and actual effect. In addition, we used $L_1$-regularization for feature selection, reducing the number of features to 328 while still achieving similar F1 score to (Hancock et al., 2018); they reported an F1 of 42.3, while we achieved 42.0. After feature selection, we remove the $L_1$-regularization and train a $L_2$-regularized logistic regression model. We assume that the feature selection step is static and not affected by removing groups of data (though in general this assumption is not true); we therefore do not include feature selection in our influence calculations.

We note that in BabbleLabble, a given LF can never output positive on one example but negative on another. Hence, some LFs are positive (unable to output negative and only able to abstain or output positive), while the others are negative (unable to output positive and only able to abstain or output negative).

## B.2 MultiNLI

Williams et al. (2018) created the MultiNLI dataset for the task of natural language inference: determining if a pair of sentences agree, contradict, or are neutral. To do so, they presented crowdworkers with initial sentences and asked them to generate follow-on sentences that were neutral or in agreement/contradiction. For example, a crowdworker may be presented with "Met my first girlfriend that way." and write the contradicting sentence "I didn't meet my first girlfriend until later." (Williams et al., 2018). Thus, each of the 380 crowdworkers generated a subset of the dataset. We used these subsets in our application of influence.

The training set consisted of 392,702 examples from five genres. The development set consisted of 10,000 "matched" examples from the same five genres as the training set, as well as 10,000 "mismatched" examples from five new genres. The test set was put on Kaggle as an open competition, hence we do not have its labels and could not use it; therefore, we use the development set as the test set.

The continuous bag-of-words baseline in Williams et al. (2018) first converted the raw text of each sentence in the pair into a vector by treating the sentence as a continuous bag of words and simply averaging the 300-dimensional GloVe vector embeddings. This converted a pair of sentences into vectors $a, b$. They then concatenated $[a, b, a - b, a \odot b]$ into a 1200D vector, where $a \odot b$ denotes the element-wise product. Finally, they treated this as input to a neural network with three hidden layers and fine-tuned the entire model, including word embeddings (more details in (Williams et al., 2018)).

For our application, we truncated their baseline and just used the concatenation of $a$ and $b$ as the representation for every example. By running logistic regression on this, we achieved test accuracy of $50.4\%$ (vs. their baseline's $64.7\%$; the performance difference comes from the additional dimensions in their vector embeddings and the finetuning through the neural network). Future work could explore influence in the setting of more complex and higher-performing models.

# C  Additional experiments

As in Figure 1, in each of the plots below, the grey reference line has slope 1, and the red borders represent points that are not plotted because they are outside the x- or y-axis range.

## C.1  Representative test points

Figure C.1 is similar to Figure 1 in the main text: it shows the influences vs. actual effects of groups on test points, but with test points that are closer to the median (within the 40th to 60th percentile) of the test loss distribution.

## C.2  Regularization

In Section 4, our bounds show that influence ought to be closer to actual effect as regularization increases. Here, we support this claim empirically on Diabetes, Enron, Dogfish, and MNIST (small).[5] To do so, for each dataset, we selected a range of values for $\lambda/n$, and we selected subsets as described

Figure C.1: Influences vs. actual effects of the same coherent groups in Figure 1, but on test points closer to the median (within the 40th to 60th percentile) of the test loss distribution. We consider these to represent average test points. On these, influence on the test prediction remains well-correlated with the actual effect.

Figure C.2: The effect of regularization for a representative test point. Red frame lines indicate the existence of points exceeding those bounds. We did not include the test prediction for MNIST (small) because the margin is not well-defined for a multiclass model.

in Appendix A.2. We then computed the influence and actual effect of each of these subsets on a representative test point's prediction, that point's loss, and on self-loss (Figure C.2).

Figure C.3: As regularization increases, correlation increases between the influences and actual effects on test prediction (Left), test loss (Middle), and self-loss (Right).

In Figure C.3, we observe the trend that correlation generally increases as $\lambda$ does. Specifically, we computed the Spearman $\rho$ between the influence and actual effect for each dataset, each value of $\lambda$, and each evaluation function $f(\cdot)$ of interest (i.e., test prediction, test loss, or self-loss).

## C.3 The effect of loss curvature on the accuracy of influence

One takeaway from the results on test loss in Figure 1-Mid is that the curvature of $f(\theta)$ can significantly increase approximation error; this is expected since the influence $\mathcal{I}_f(w)$ linearizes $f(\cdot)$ around $\hat{\theta}(\mathbf{1})$. When possible, choosing a $f(\cdot)$ that has low curvature (e.g., the linear prediction) will result in higher accuracy. We can mitigate this by using influence to approximate the parameters $\hat{\theta}(\mathbf{1} - w)$ and then plug that estimate into $f(\cdot)$ (Figure C.4), though this can be more computationally expensive.

Note that Figure C.4 shows that this technique does not help much for measuring self-loss. However, in the context of LOOCV, the computational complexity of the Newton approximation for self-loss (described in Section 4) is similar to that of the influence approximation, so we encourage the use of the Newton approximation for LOOCV (as in Rad and Maleki (2018)); Figure C.4 shows that this leads to more accurate approximations for self-loss.

## C.4 Additional analysis of influence functions applied to natural groups of data

In Section 5, we considered the CDR and MultiNLI datasets, which contain the natural subsets of LFs and crowdworkers, respectively. To draw inferences about these subsets, we took the $L_2$-regularized logistic regression model described in Appendix A, calculated the influence of the LF/crowdworker subsets, and retrained the model once for each LF/crowdworker.

**CDR.**  As discussed in Appendix B.1, an LF is either positive or negative, where a positive LF can only give positive labels or abstain, and similarly for negative LFs. Because of this stark class separation, we indicate whether an LF is positive or negative, and we consider LF influence on the positive test examples separately from their influence on the negative test examples. To measure an LF's influence and actual effect on a set of test points, we simply add up its influence and actual effect on the set's individual test points.

In Figure C.5, we note that influence is a good approximation of an LF's actual effect, just as with other kinds of subsets as well as other datasets (Figure 1). Furthermore, we observe that positive LFs improve the overall performance of the positively labeled portion of the test set while hurting the negatively labeled portion of the test set, and vice versa for negative LFs. This dichotomous effect further motivates the analysis of influence on the positive test set separately from the negative test set, since the process of adding these two influences to study the influence on the entire test set would obscure the full story.

Figure C.4: The top 3 rows show influence on test loss (with the same test points as in Figure 1), while the bottom 3 show self-loss. Within each set, the first row shows the influence vs. actual effect (as in Figure 1); the second shows the predicted effect obtained by estimating the change in parameters via influence and then evaluating $f(\cdot)$ directly on those parameters; and the third shows the Newton approximation.

Next, we define an LF's *coverage* to be the proportion of the examples that it does not abstain on, which can be measured through the number of examples in its corresponding subset. In Figure C.6, we observe that the magnitude of influence correlates strongly with coverage.

Finally, we define an LF's *precision* to be the number of examples it labels correctly divided by the number of examples it does not abstain on. Because the dataset had many more negative than positive examples, positive LFs had lower precision than negative LFs. Surprisingly, even when this effect

Figure C.5: We observe both correlation and underestimation for LFs on the positive and negative test sets. We also see that positive LFs help the positive test set and hurt the negative test set; vice versa for negative LFs.

Figure C.6: The magnitude of LF influence correlates with coverage. This figure is an extension of Figure 4-Left: there, we showed the influence of positive LFs on the positive test set and the influence of negative LFs on the negative test set. Here, we additionally show the influence of positive LFs on the negative test set and vice versa.

was taken into account and we considered positive LFs separately from negative ones, precision did not correlate with influence (Figure C.7).

Figure C.7: LF influence does not correlate with precision. Similar to Figure C.6, this figure is an extension of Figure 4-Mid: there, we showed the influence of positive LFs on the positive test set and the influence of negative LFs on the negative test set. Here, we additionally show the influence of positive LFs on the negative test set and vice versa.

**MultiNLI.**   As discussed in Appendix B.2, the training set consisted of five genres, and the test set consisted of a matched portion with the same five genres, as well as a mismatched portion with five new genres. For succinctness, we refer to the influence/actual effect of the set of examples generated by a single crowdworker as that crowdworker's influence/actual effect.

First, we note in Figure C.8 that influence is a good approximation of a crowdworker's actual effect for both matched and mismatched test sets, consistent with our findings in Figure 1 for other subset types and datasets.

Figure C.8: We observe strong correlation between crowdworkers' influence and actual effects.

Unlike in CDR (Figure C.6), we do not find strong correlation between a crowdworker's influence and the number of examples they contributed; it is possible to contribute many examples but have relatively little influence (Figure C.9).

Figure C.9: Size does not correlate strongly with influence. The hidden point is the crowdworker that contributed 35,000 examples. This is the figure presented in Figure 4-Right.

The most prolific crowdworker contributed 35,000 examples and had large negative influence on the test set. A closer analysis revealed that they had positive influence on the fiction genre but lowered performance on many other genres, despite contributing roughly equally to each genre. This genre-specific trend tended to hold more broadly among the workers: there appear to be two categories of genres (fiction, facetoface, nineeleven vs. travel, government, verbatim, letters, oup)

such that each worker tended to have positive influence on all genres in one category and negative influence on all genres in the other (Figure C.10). Moreover, the number of examples a worker contributed to a given genre was not a good indicator for their influence on that genre (Figure C.11).

Figure C.10: Workers tended to have positive influence on fiction, facetoface, and nineeleven and negative influence on travel, government, verbatim, letters, and oup (or vice versa). In this plot, we allowed for full color saturation when the magnitude of the total influence on the test set (matched) exceeded 0.8.

Figure C.11: Influence on a genre does not correlate with number of contributions to that genre.

# D    Additional analysis on influence vs. actual effect on a test point

## D.1    Counterexamples

For Figure 3, we constructed two binary datasets in which the influence of a certain class of subsets on the test prediction of a single test point exhibits pathological behavior.

**Rotation effect.**    In Figure 3-Left, our aim was to show that there can be a dataset with subsets such that the cone constraint discussed in Section 4.4 does not hold.

The rotation effect described in Corollary 1 is due to the angular difference between the change in parameters predicted by the influence approximation, $\Delta\theta_{\mathrm{inf}}(w) \overset{\text{def}}{=} H_{\lambda,\mathbf{1}}^{-1}g_{\mathbf{1}}(w) = H_{\lambda,\mathbf{1}}^{-\frac{1}{2}}v_w$, and the change in parameters predicted by the Newton approximation, $\Delta\theta_{\mathrm{Nt}}(w) = H_{\lambda,\mathbf{1}}^{-\frac{1}{2}}\big(D(w) + I\big)v_w$. If $\Delta\theta_{\mathrm{inf}}(w)$ and $\Delta\theta_{\mathrm{Nt}}(w)$ are linearly independent, then for any pair of target values $a, b \in \mathbb{R}$, we can find some $x_{\mathrm{test}}$ such that $\mathcal{I}_f(w) = x_{\mathrm{test}}{}^\top\Delta\theta_{\mathrm{inf}}(w) = a$ and $\mathcal{I}_f^{\mathrm{Nt}}(w = x_{\mathrm{test}}{}^\top\Delta\theta_{\mathrm{Nt}}(w) = b$.

To exploit this, we constructed the *MoG* dataset as an equal mixture of two standard (identity covariance) Gaussian distributions in $\mathbb{R}^{60}$, one for each class, and with means $(-1/2, 0, \ldots, 0)$ and $(1/2, 0, \ldots, 0)$, respectively. In particular:

1. We sampled 60 examples from each class for a total of $n = 120$ training points, and set the regularization strength $\lambda = 0.001$.

2. We then computed $\Delta\theta_{\mathrm{inf}}(w)$ and $\Delta\theta_{\mathrm{Nt}}(w)$ for each pair of training points and chose the 120 pairs of training points with the largest angles between $\Delta\theta_{\mathrm{inf}}(w)$ and $\Delta\theta_{\mathrm{Nt}}(w)$.

3. Finally, we solved a least-squares optimization problem to find $x_{\mathrm{test}}$ for which $\mathcal{I}_f(w)$ and $\mathcal{I}_f^{\mathrm{Nt}}(w)$ are approximately decorrelated.

Note that we adversarially chose which subsets to study in this counterexample, since our main goal was to show that there existed subsets for which the cone constraint did not hold. For the next counterexample, we instead study all possible subsets in the restricted setting of removing copies of single points.

**Scaling effect.** In Figure 3-Right, our aim was to construct a dataset such that even if we only removed subsets comprising copies of single distinct points, a low influence need not translate into a low actual effect.

To do so, we constructed the *Ortho* dataset that contains 2 repeated points of opposite classes on each of the 2 canonical axes of $\mathbb{R}^2$ (for a total of 4 distinct points). By varying their relative distances from the origin, we can control the influence of removing one of these points as well as the rate that the scaling factor $d(w)$ from Proposition 4 grows as we remove more copies of the same point. Furthermore, because the axes are orthogonal, we can control $d(w)$ independently for each repeated point. We fix the test point $x_{\mathrm{test}} = (1, 1)$. Maximizing $d(w)$ for one axis and minimizing it for the other produces the two distinct lines in Figure 3-Right.

### D.2 Scaling effects when removing multiple points

In the general setting of removing subsets of different points, the analogous failure case to a varying scaling factor $d(w)$ (Figure 3-Right) is the varying scaling effect that the error matrix $D(w)$ in Proposition 2 can have on different subsets $w$. The range of this effect is bounded by the spectral norm of $D(w)$. This norm is precisely equal to $d(w)$ in the single-point setting, and it is large when we remove a subset $w$ whose Hessian $H_{\mathbf{1}}(w)$ is almost as large as the full Hessian $H_{\lambda,\mathbf{1}}$ in some direction. As with $d(w)$, the spectral norm of $D(w)$ decreases with $\lambda$ (Proposition 2), so as regularization increases, we expect that the influence of a group will track its actual effect more accurately.

### D.3 The relationship between influence and actual effect on the loss of a test point

In the margin-based setting, the loss $\ell(x_{\mathrm{test}}, y_{\mathrm{test}}; \theta)$ is a monotone function of the linear prediction $\theta^\top x_{\mathrm{test}}$. Thus, measuring $f(\theta) = \ell(x_{\mathrm{test}}, y_{\mathrm{test}}; \theta)$ will display the same rank correlation as measuring $f(\theta) = \theta^\top x_{\mathrm{test}}$ above, so the same results about correlation in the test prediction setting carry over.

However, the second-order $f$-curvature term $\frac{1}{2}\Delta\theta_{\mathrm{Nt}}(w)^\top \nabla_\theta^2 f(\hat{\theta}(\mathbf{1}))\Delta\theta_{\mathrm{Nt}}(w)$ from Proposition 2 is always non-negative, even if the influence is negative. Under the assumption that $\mathrm{Err}_{f,3}(w)$ and $\mathrm{Err}_{\mathrm{Nt\text{-}act}}(w)$ are both small because they decay as $O(1/\lambda^3)$, this implies that underestimation is only preserved when the influence is positive, as we observed empirically in Figure 1-Mid.

## E Proofs

### E.1 Notation

We first review the notation given in Section 2 and introduce new definitions that will be useful in the sequel. We define the empirical risk as

$$L_s(\theta) \stackrel{\mathrm{def}}{=} \left[\sum_{i=1}^{n} s_i \ell(x_i, y_i; \theta)\right] + \frac{\lambda}{2}\|\theta\|_2^2,$$

such that the optimal parameters are $\hat{\theta}(s) \stackrel{\mathrm{def}}{=} \arg\min_{\theta\in\Theta} L_s(\theta)$.

Given sample weight vectors $r, s \in \mathbb{R}^n$, we define the derivatives

$$g_r(s) \stackrel{\text{def}}{=} \sum_{i=1}^{n} s_i \nabla_\theta \ell(x_i, y_i; \hat{\theta}(r))$$

$$H_r(s) \stackrel{\text{def}}{=} \sum_{i=1}^{n} s_i \nabla_\theta^2 \ell(x_i, y_i; \hat{\theta}(r)).$$

If the argument $s$ is omitted, it is assumed to be equal to $r$. For example,

$$H_{\mathbf{1}} \stackrel{\text{def}}{=} \sum_{i=1}^{n} \nabla_\theta^2 \ell(x_i, y_i; \hat{\theta}(\mathbf{1})).$$

If $H$ has a $\lambda$ subscript, then we add $\lambda I$. For example,

$$H_{\lambda,\mathbf{1}} \stackrel{\text{def}}{=} H_{\mathbf{1}} + \lambda I.$$

For a given dataset, we define the following constants:

$$C_\ell = \max_{1 \leq i \leq n} \left\| \nabla_\theta \ell(x_i, y_i, \hat{\theta}(\mathbf{1})) \right\|_2,$$
$$\sigma_{\min} = \text{smallest singular value of } H_{\mathbf{1}},$$
$$\sigma_{\max} = \text{largest singular value of } H_{\mathbf{1}}.$$

To avoid confusion with the vector 2-norm, we will use the operator norm $\|\cdot\|_{\text{op}}$ to denote the matrix 2-norm.

In the sequel, we study the order-3 tensor $\nabla_\theta^3 f(\hat{\theta}(\mathbf{1}))$. We define its product with a vector (which returns a matrix) as a contraction along the last dimension:

$$\left\langle \nabla_\theta^3 f(\hat{\theta}(\mathbf{1})), v \right\rangle_{ij} \stackrel{\text{def}}{=} \sum_k \frac{\partial^3 f(\hat{\theta}(\mathbf{1}))}{\partial \theta_i \partial \theta_j \partial \theta_k} v_k.$$

## E.2 Assumptions

We make the following assumptions on the derivatives of the loss $\ell(x, y, \theta)$ and the evaluation function $f(\theta)$.

**Assumption 1** (Positive-definiteness and Lipschitz continuity of $H$)**.** *The loss $\ell(x, y, \theta)$ is convex and twice-differentiable in $\theta$, with positive regularization $\lambda > 0$. Further, there exists $C_H \in \mathbb{R}$ such that*

$$\left\| \nabla_\theta^2 \ell(x, y, \theta_1) - \nabla_\theta^2 \ell(x, y, \theta_2) \right\|_{\text{op}} \leq C_H \|\theta_1 - \theta_2\|_2$$

*for all $(x, y) \in \mathcal{X} \times \mathcal{Y}$ and $\theta_1, \theta_2 \in \Theta$. This is a bound on the third derivative of $\ell$, if it exists.*

**Assumption 2** (Bounded derivatives of $f$)**.** *$f(\theta)$ is thrice-differentiable, with $C_f, C_{f,3} \in \mathbb{R}$ such that*

$$C_f = \sup_{\theta \in \Theta} \|\nabla_\theta f(\theta)\|_2, \qquad C_{f,3} = \sup_{v \in \Theta, \|v\|_2 = 1} \left\| \left\langle \nabla_\theta^3 f(\hat{\theta}(\mathbf{1})), v \right\rangle \right\|_{\text{op}}.$$

These assumptions apply to all the results that follow below.

## E.3 Bounding the error of the one-step Newton approximation

**Proposition 1** (Restated)**.** *Let the Newton error be $\text{Err}_{\text{Nt-act}}(w) \stackrel{\text{def}}{=} \mathcal{I}_f^*(w) - \mathcal{I}_f^{\text{Nt}}(w)$. Then under Assumptions 1 and 2,*

$$|\text{Err}_{\text{Nt-act}}(w)| \leq \frac{n\|w\|_1^2 C_f C_H C_\ell^2}{(\sigma_{\min} + \lambda)^3}.$$

$\text{Err}_{\text{Nt-act}}(w)$ *only involves third-order or higher derivatives of the loss, so it is 0 for quadratic losses.*

*Proof.* This proof is adapted to our setting from the standard analysis of the Newton method in convex optimization (Boyd and Vandenberghe, 2004).

First, note that $\mathrm{Err}_{\text{Nt-act}}(w) = \mathcal{I}_f^*(w) - \mathcal{I}_f^{\text{Nt}}(w) = f(\hat{\theta}(\mathbf{1} - w)) - f(\hat{\theta}_{\text{Nt}}(\mathbf{1} - w))$. We will bound the norm of the difference of the parameters $\left\| \hat{\theta}(\mathbf{1} - w) - \hat{\theta}_{\text{Nt}}(\mathbf{1} - w) \right\|_2$; the desired bound on $f$ then follows from the assumption that $f$ has gradients bounded by $C_f$ and is therefore Lipschitz.

Since $L_{\mathbf{1}-w}(\theta)$ is strongly convex (with parameter $\sigma_{\min} + \lambda$) and minimized by $\hat{\theta}(\mathbf{1} - w)$, we can bound the distance $\left\| \hat{\theta}(\mathbf{1} - w) - \hat{\theta}_{\text{Nt}}(\mathbf{1} - w) \right\|_2$ in terms of the norm of the gradient at $\hat{\theta}_{\text{Nt}}(\mathbf{1} - w)$:

$$\left\| \hat{\theta}(\mathbf{1} - w) - \hat{\theta}_{\text{Nt}}(\mathbf{1} - w) \right\|_2 \leq \frac{2}{\sigma_{\min} + \lambda} \left\| \nabla_\theta L_{\mathbf{1}-w} \left( \hat{\theta}_{\text{Nt}}(\mathbf{1} - w) \right) \right\|_2.$$

Therefore, the problem reduces to bounding $\left\| \nabla_\theta L_{\mathbf{1}-w} \left( \hat{\theta}_{\text{Nt}}(\mathbf{1} - w) \right) \right\|_2$.

We start by expressing the Newton step $\Delta\theta_{\text{Nt}}(w)$ in terms of the first and second derivatives of the empirical risk $L_{\mathbf{1}-w}(\theta)$:

$$g_{\mathbf{1}}(w) = \sum_{i=1}^{n} w_i \nabla_\theta \ell(x_i, y_i; \hat{\theta}(\mathbf{1}))$$
$$= -\sum_{i=1}^{n} (1 - w_i) \nabla_\theta \ell(x_i, y_i; \hat{\theta}(\mathbf{1}))$$
$$= -\nabla_\theta L_{\mathbf{1}-w}(\hat{\theta}(\mathbf{1})),$$
$$H_{\lambda,\mathbf{1}}(\mathbf{1} - w) = \sum_{i=1}^{n} (1 - w_i) \nabla_\theta^2 \ell(x_i, y_i; \hat{\theta}(\mathbf{1}))$$
$$= \nabla_\theta^2 L_{\mathbf{1}-w}(\hat{\theta}(\mathbf{1})),$$
$$\Delta\theta_{\text{Nt}}(w) = H_{\lambda,\mathbf{1}}(\mathbf{1} - w)^{-1} g_{\mathbf{1}}(w)$$
$$= -\left[ \nabla_\theta^2 L_{\mathbf{1}-w}(\hat{\theta}(\mathbf{1})) \right]^{-1} \nabla_\theta L_{\mathbf{1}-w}(\hat{\theta}(\mathbf{1})),$$

where the second equality for $g_{\mathbf{1}}(w)$ comes from the fact that at the optimum $\hat{\theta}(\mathbf{1})$, the sum of the gradients $\sum_{i=1}^{n} \nabla_\theta \ell(x_i, y_i; \hat{\theta}(\mathbf{1}))$ is 0.

With these expressions, we bound the norm of the gradient $\nabla_\theta L_{\mathbf{1}-w}(\hat{\theta}_{\text{Nt}}(\mathbf{1} - w))$:

$$\left\| \nabla_\theta L_{\mathbf{1}-w} \left( \hat{\theta}_{\text{Nt}}(\mathbf{1} - w) \right) \right\|_2$$
$$= \left\| \nabla_\theta L_{\mathbf{1}-w} \left( \hat{\theta}(\mathbf{1}) + \Delta\theta_{\text{Nt}}(w) \right) \right\|_2$$
$$= \left\| \nabla_\theta L_{\mathbf{1}-w} \left( \hat{\theta}(\mathbf{1}) + \Delta\theta_{\text{Nt}}(w) \right) - \nabla_\theta L_{\mathbf{1}-w} \left( \hat{\theta}(\mathbf{1}) \right) - \nabla_\theta^2 L_{\mathbf{1}-w} \left( \hat{\theta}(\mathbf{1}) \right) \Delta\theta_{\text{Nt}}(w) \right\|_2$$
$$= \left\| \int_0^1 \left( \nabla_\theta^2 L_{\mathbf{1}-w} \left( \hat{\theta}(\mathbf{1}) + t\Delta\theta_{\text{Nt}}(w) \right) - \nabla_\theta^2 L_{\mathbf{1}-w} \left( \hat{\theta}(\mathbf{1}) \right) \right) \Delta\theta_{\text{Nt}}(w) \, dt \right\|_2$$
$$\leq \frac{nC_H}{2} \left\| \Delta\theta_{\text{Nt}}(w) \right\|_2^2$$
$$= \frac{nC_H}{2} \left\| \left[ \nabla_\theta^2 L_{\mathbf{1}-w}(\hat{\theta}(\mathbf{1})) \right]^{-1} \nabla_\theta L_{\mathbf{1}-w}(\hat{\theta}(\mathbf{1})) \right\|_2^2$$
$$\leq \frac{nC_H}{2(\sigma_{\min} + \lambda)^2} \left\| \nabla_\theta L_{\mathbf{1}-w}(\hat{\theta}(\mathbf{1})) \right\|_2^2$$
$$\leq \frac{n \left\| w \right\|_1^2 C_H C_\ell^2}{2(\sigma_{\min} + \lambda)^2}.$$

Putting together the successive bounds gives the result. $\square$

### E.4 Characterizing the difference between the Newton approximation and influence

Before proving Proposition 2, we first prove a lemma about the spectrum of the error matrix $D(w)$.

**Lemma 1.** *The matrix* $D(w) \stackrel{\text{def}}{=} \left(I - H_{\lambda,\mathbf{1}}^{-\frac{1}{2}} H_{\mathbf{1}}(w) H_{\lambda,\mathbf{1}}^{-\frac{1}{2}}\right)^{-1} - I$ *has singular values bounded between 0 and* $\frac{\sigma_{\max}}{\lambda}$.

*Proof.* We first show that $H_{\lambda,\mathbf{1}}^{-\frac{1}{2}} H_{\mathbf{1}}(w) H_{\lambda,\mathbf{1}}^{-\frac{1}{2}}$ has singular values bounded between 0 and $\frac{\sigma_{\max}}{\sigma_{\max}+\lambda}$. The lower bound of 0 comes from the fact that $H_{\lambda,\mathbf{1}}^{-\frac{1}{2}} H_{\mathbf{1}}(w) H_{\lambda,\mathbf{1}}^{-\frac{1}{2}}$ is symmetric and $H_{\mathbf{1}}(w) \succeq 0$.

To show the upper bound, first note that $H_{\mathbf{1}}(w) \preceq H_{\mathbf{1}}(w) + H_{\mathbf{1}}(\mathbf{1} - w) = H_{\mathbf{1}}$ (recalling that $w \in \{0,1\}^n$), and let $U\Sigma U^\top$ be the singular value decomposition of $H_{\mathbf{1}}$. Since $H_{\lambda,\mathbf{1}} = H_{\mathbf{1}} + \lambda I$, we have

$$
\begin{aligned}
H_{\lambda,\mathbf{1}}^{-\frac{1}{2}} H_{\mathbf{1}}(w) H_{\lambda,\mathbf{1}}^{-\frac{1}{2}} &= \left(H_{\mathbf{1}} + \lambda I\right)^{-\frac{1}{2}} H_{\mathbf{1}}(w) \left(H_{\mathbf{1}} + \lambda I\right)^{-\frac{1}{2}} \\
&\preceq \left(H_{\mathbf{1}} + \lambda I\right)^{-\frac{1}{2}} H_{\mathbf{1}} \left(H_{\mathbf{1}} + \lambda I\right)^{-\frac{1}{2}} \\
&= \left(U(\Sigma + \lambda I)U^\top\right)^{-\frac{1}{2}} U\Sigma U^\top \left(U(\Sigma + \lambda I)U^\top\right)^{-\frac{1}{2}} \\
&= U(\Sigma + \lambda I)^{-\frac{1}{2}} \Sigma (\Sigma + \lambda I)^{-\frac{1}{2}} U^\top,
\end{aligned}
$$

so its maximum singular value is upper bounded by $\frac{\sigma_{\max}}{\sigma_{\max}+\lambda}$.

The bound on the singular values of $H_{\lambda,\mathbf{1}}^{-\frac{1}{2}} H_{\mathbf{1}}(w) H_{\lambda,\mathbf{1}}^{-\frac{1}{2}}$ implies that the singular values of $I - H_{\lambda,\mathbf{1}}^{-\frac{1}{2}} H_{\mathbf{1}}(w) H_{\lambda,\mathbf{1}}^{-\frac{1}{2}}$ lie in $\left[\frac{\lambda}{\sigma_{\max}+\lambda}, 1\right]$. In turn, this implies that the singular values of $\left(I - H_{\lambda,\mathbf{1}}^{-\frac{1}{2}} H_{\mathbf{1}}(w) H_{\lambda,\mathbf{1}}^{-\frac{1}{2}}\right)^{-1}$ lie in $\left[1, \frac{\sigma_{\max}+\lambda}{\lambda}\right]$. Subtracting 1 from each end (for the identity matrix) gives the desired result. $\square$

**Proposition 2** (Restated). *Under Assumptions 1 and 2, the Newton-influence error* $\mathrm{Err}_{\text{Nt-inf}}(w)$ *is*

$$
\mathrm{Err}_{\text{Nt-inf}}(w) = \nabla_\theta f(\hat{\theta}(\mathbf{1}))^\top H_{\lambda,\mathbf{1}}^{-\frac{1}{2}} D(w) H_{\lambda,\mathbf{1}}^{-\frac{1}{2}} g_{\mathbf{1}}(w) \; + \; \underbrace{\frac{1}{2}\Delta\theta_{\text{Nt}}(w)^\top \nabla_\theta^2 f(\hat{\theta}(\mathbf{1}))\Delta\theta_{\text{Nt}}(w) + \mathrm{Err}_{f,3}(w)}_{\textit{Error from curvature of } f(\cdot)},
$$

*with* $D(w) \stackrel{\text{def}}{=} \left(I - H_{\lambda,\mathbf{1}}^{-\frac{1}{2}} H_{\mathbf{1}}(w) H_{\lambda,\mathbf{1}}^{-\frac{1}{2}}\right)^{-1} - I$ *and* $H_{\mathbf{1}}(w) \stackrel{\text{def}}{=} \sum_{i=1}^{n} w_i \nabla_\theta^2 \ell(x_i, y_i; \hat{\theta}(\mathbf{1}))$. *The* error matrix $D(w)$ *has eigenvalues between 0 and* $\frac{\sigma_{\max}}{\lambda}$, *where* $\sigma_{\max}$ *is the largest eigenvalue of* $H_{\mathbf{1}}$. *The residual term* $\mathrm{Err}_{f,3}(w)$ *captures the error due to third-order derivatives of* $f(\cdot)$ *and is bounded by* $|\mathrm{Err}_{f,3}(w)| \leq \|w\|_1^3 C_{f,3} C_\ell^3 / 6(\sigma_{\min} + \lambda)^3$.

*Proof.* From the second-order Taylor expansion of $f$ about $\hat{\theta}(\mathbf{1})$, there exists $0 \leq \xi \leq 1$ such that

$$
\begin{aligned}
\mathcal{I}_f^{\text{Nt}}(w) &= f(\hat{\theta}_{\text{Nt}}(\mathbf{1} - w)) - f(\hat{\theta}(\mathbf{1})) \\
&= f(\hat{\theta}(\mathbf{1}) + \Delta\theta_{\text{Nt}}(w)) - f(\hat{\theta}(\mathbf{1})) \\
&= \nabla_\theta f(\hat{\theta}(\mathbf{1}))^\top \Delta\theta_{\text{Nt}}(w) + \frac{1}{2}\Delta\theta_{\text{Nt}}(w)^\top \nabla_\theta^2 f(\hat{\theta}(\mathbf{1}))\Delta\theta_{\text{Nt}}(w) + \\
&\quad \frac{1}{6}\Delta\theta_{\text{Nt}}(w)^\top \left\langle \nabla_\theta^3 f(\hat{\theta}(\mathbf{1}) + \xi\Delta\theta_{\text{Nt}}(w)), \Delta\theta_{\text{Nt}}(w)\right\rangle \Delta\theta_{\text{Nt}}(w) \\
&= \nabla_\theta f(\hat{\theta}(\mathbf{1}))^\top \Delta\theta_{\text{Nt}}(w) + \frac{1}{2}\Delta\theta_{\text{Nt}}(w)^\top \nabla_\theta^2 f(\hat{\theta}(\mathbf{1}))\Delta\theta_{\text{Nt}}(w) + \mathrm{Err}_{f,3}(w), \quad (6)
\end{aligned}
$$

where we define $\mathrm{Err}_{f,3}(w) \stackrel{\text{def}}{=} \frac{1}{6}\Delta\theta_{\text{Nt}}(w)^\top \left\langle \nabla_\theta^3 f(\hat{\theta}(\mathbf{1}) + \xi\Delta\theta_{\text{Nt}}(w)), \Delta\theta_{\text{Nt}}(w)\right\rangle \Delta\theta_{\text{Nt}}(w)$ to be the error due to third-order and higher derivatives of $f$.

We can express the difference between the first-order Taylor term $\nabla_\theta f(\hat{\theta}(\mathbf{1}))^\top \Delta\theta_{\mathrm{Nt}}(w)$ above and the first-order influence approximation $\mathcal{I}_f(w) = q'_w(0) = \nabla_\theta f(\hat{\theta}(\mathbf{1}))^\top H_{\lambda,\mathbf{1}}^{-1} g_{\mathbf{1}}(w)$ as

$$
\begin{aligned}
&\nabla_\theta f(\hat{\theta}(\mathbf{1}))^\top \Delta\theta_{\mathrm{Nt}}(w) - \mathcal{I}_f(w) \\
&= \nabla_\theta f(\hat{\theta}(\mathbf{1}))^\top \Delta\theta_{\mathrm{Nt}}(w) - \nabla_\theta f(\hat{\theta}(\mathbf{1}))^\top H_{\lambda,\mathbf{1}}^{-1} g_{\mathbf{1}}(w) \\
&= \nabla_\theta f(\hat{\theta}(\mathbf{1}))^\top \left( H_{\lambda,\mathbf{1}}(\mathbf{1}-w)^{-1} - H_{\lambda,\mathbf{1}}^{-1} \right) g_{\mathbf{1}}(w) \\
&= \nabla_\theta f(\hat{\theta}(\mathbf{1}))^\top \left( \left( H_{\lambda,\mathbf{1}} - H_{\mathbf{1}}(w) \right)^{-1} - H_{\lambda,\mathbf{1}}^{-1} \right) g_{\mathbf{1}}(w) \\
&= \nabla_\theta f(\hat{\theta}(\mathbf{1}))^\top H_{\lambda,\mathbf{1}}^{-\frac{1}{2}} \left( H_{\lambda,\mathbf{1}}^{\frac{1}{2}} \left( H_{\lambda,\mathbf{1}} - H_{\mathbf{1}}(w) \right)^{-1} H_{\lambda,\mathbf{1}}^{\frac{1}{2}} - I \right) H_{\lambda,\mathbf{1}}^{-\frac{1}{2}} g_{\mathbf{1}}(w) \\
&= \nabla_\theta f(\hat{\theta}(\mathbf{1}))^\top H_{\lambda,\mathbf{1}}^{-\frac{1}{2}} \left( \left( I - H_{\lambda,\mathbf{1}}^{-\frac{1}{2}} H_{\mathbf{1}}(w) H_{\lambda,\mathbf{1}}^{-\frac{1}{2}} \right)^{-1} - I \right) H_{\lambda,\mathbf{1}}^{-\frac{1}{2}} g_{\mathbf{1}}(w) \\
&= \nabla_\theta f(\hat{\theta}(\mathbf{1}))^\top H_{\lambda,\mathbf{1}}^{-\frac{1}{2}} D(w) H_{\lambda,\mathbf{1}}^{-\frac{1}{2}} g_{\mathbf{1}}(w). \tag{7}
\end{aligned}
$$

Substituting (7) into (6), we have that

$$
\begin{aligned}
\mathcal{I}_f^{\mathrm{Nt}}(w) - \mathcal{I}_f(w) &= \nabla_\theta f(\hat{\theta}(\mathbf{1}))^\top H_{\lambda,\mathbf{1}}^{-\frac{1}{2}} D(w) H_{\lambda,\mathbf{1}}^{-\frac{1}{2}} g_{\mathbf{1}}(w) \\
&\quad + \frac{1}{2} \Delta\theta_{\mathrm{Nt}}(w)^\top \nabla_\theta^2 f(\hat{\theta}(\mathbf{1})) \Delta\theta_{\mathrm{Nt}}(w) + \mathrm{Err}_{f,3}(w),
\end{aligned}
$$

as desired.

We can bound $\mathrm{Err}_{f,3}(w)$ as follows:

$$
\begin{aligned}
|\mathrm{Err}_{f,3}(w)| &\leq \frac{C_{f,3}}{6} \|\Delta\theta_{\mathrm{Nt}}(w)\|_2^3 \\
&\leq \frac{\|w\|_1^3 \, C_{f,3} C_\ell^3}{6(\sigma_{\min} + \lambda)^3}.
\end{aligned}
$$

Applying Lemma 1 to bound the spectrum of $D(w)$ completes the proof. □

## E.5 The influence on self-loss

We first state two linear algebra facts that will be useful in the sequel.

**Lemma 2.** *Let $A \succ 0, B \succeq 0 \in \mathbb{R}^{d \times d}$ be a pair of symmetric positive-definite and positive-semidefinite matrices, respectively. Let $\sigma_{A,1}$ be the largest eigenvalue of $A$, $\sigma_{A,d}$ the smallest eigenvalue of $A$, and similarly let $\sigma_{B,1}$ and $\sigma_{B,d}$ be the largest and smallest eigenvalues of $B$, respectively. Then*

$$
\frac{\sigma_{B,d}}{\sigma_{A,1}} I \preceq A^{-\frac{1}{2}} B A^{-\frac{1}{2}} \preceq \frac{\sigma_{B,1}}{\sigma_{A,d}} I.
$$

*Proof.* Note that $\frac{1}{\sigma_{A,1}}$ is the smallest eigenvalue of $A^{-1}$, while $\frac{1}{\sigma_{A,d}}$ is its largest. The lemma follows from the fact that the smallest singular value of the product of two matrices is lower bounded by the product of the smallest singular values of each matrix, and similarly the largest singular value of the product is upper bounded by the product of the largest singular values of each matrix. □

The next fact is a consequence of the variational definition of eigenvalues.

**Lemma 3.** *Given a symmetric matrix $A \in \mathbb{R}^{d \times d}$ and a vector $v \in \mathbb{R}^d$, we have the following bounds on the quadratic form $v^\top A v$:*

$$
\sigma_d \|v\|_2^2 \leq v^\top A v \leq \sigma_1 \|v\|_2^2,
$$

*where $\sigma_d$ is the smallest eigenvalue of $A$, and $\sigma_1$ is the largest.*

We are now ready to analyze the effect of removing a subset $w$ of $k$ training points on the total loss on those $k$ points.

**Proposition 3** (Restated). *Under Assumptions 1 and 2, the influence on the self-loss $f(\theta) = \sum_{i=1}^{n} w_i \ell(x_i, y_i; \theta)$ obeys*

$$\mathcal{I}_f(w) + \mathrm{Err}_{f,3}(w) \leq \mathcal{I}_f^{\mathrm{Nt}}(w) \leq \left(1 + \frac{3\sigma_{\max}}{2\lambda} + \frac{\sigma_{\max}^2}{2\lambda^2}\right) \mathcal{I}_f(w) + \mathrm{Err}_{f,3}(w).$$

*Proof.* Since $f(\theta) = \sum_{i=1}^{n} w_i \ell(x_i, y_i; \theta)$, we have that

$$\nabla_\theta f(\hat\theta(\mathbf{1})) = \sum_{i=1}^{n} w_i \nabla_\theta \ell(x_i, y_i; \hat\theta(\mathbf{1}))$$
$$= g_\mathbf{1}(w),$$
$$\nabla_\theta^2 f(\hat\theta(\mathbf{1})) = \sum_{i=1}^{n} w_i \nabla_\theta^2 \ell(x_i, y_i; \hat\theta(\mathbf{1}))$$
$$= H_\mathbf{1}(w).$$

Substituting these and $\Delta\theta_{\mathrm{Nt}}(w) = H_{\lambda,\mathbf{1}}(\mathbf{1} - w)^{-1} g_\mathbf{1}(w)$ into Proposition 2, we obtain

$$\mathcal{I}_f^{\mathrm{Nt}}(w) - \mathcal{I}_f(w) - \mathrm{Err}_{f,3}(w)$$
$$= \nabla_\theta f(\hat\theta(\mathbf{1}))^\top H_{\lambda,\mathbf{1}}^{-\frac{1}{2}} D(w) H_{\lambda,\mathbf{1}}^{-\frac{1}{2}} g_\mathbf{1}(w) + \frac{1}{2}\Delta\theta_{\mathrm{Nt}}(w)^\top \nabla_\theta^2 f(\hat\theta(\mathbf{1})) \Delta\theta_{\mathrm{Nt}}(w)$$
$$= g_\mathbf{1}(w)^\top H_{\lambda,\mathbf{1}}^{-\frac{1}{2}} D(w) H_{\lambda,\mathbf{1}}^{-\frac{1}{2}} g_\mathbf{1}(w) + \frac{1}{2} g_\mathbf{1}(w)^\top H_{\lambda,\mathbf{1}}(\mathbf{1}-w)^{-1} H_\mathbf{1}(w) H_{\lambda,\mathbf{1}}(\mathbf{1}-w)^{-1} g_\mathbf{1}(w)$$
$$= g_\mathbf{1}(w)^\top H_{\lambda,\mathbf{1}}^{-\frac{1}{2}} \left[ D(w) + \underbrace{\frac{1}{2} H_{\lambda,\mathbf{1}}^{\frac{1}{2}} H_{\lambda,\mathbf{1}}(\mathbf{1}-w)^{-1} H_\mathbf{1}(w) H_{\lambda,\mathbf{1}}(\mathbf{1}-w)^{-1} H_{\lambda,\mathbf{1}}^{\frac{1}{2}}}_{\stackrel{\mathrm{def}}{=} \Lambda(w)} \right] H_{\lambda,\mathbf{1}}^{-\frac{1}{2}} g_\mathbf{1}(w),$$

where $D(w) \stackrel{\mathrm{def}}{=} \left(I - H_{\lambda,\mathbf{1}}^{-\frac{1}{2}} H_\mathbf{1}(w) H_{\lambda,\mathbf{1}}^{-\frac{1}{2}}\right)^{-1} - I$ has singular values bounded between 0 and $\frac{\sigma_{\max}}{\lambda}$. From Lemma 2, $\Lambda(w)$ has singular values bounded between 0 and $\frac{\sigma_{\max}(\sigma_{\max}+\lambda)}{2\lambda^2}$.

Applying Lemma 3 and using $\mathcal{I}_f(w) = g_\mathbf{1}(w)^\top H_{\lambda,\mathbf{1}}^{-1} g_\mathbf{1}(w)$, we obtain

$$0 \leq g_\mathbf{1}(w)^\top H_{\lambda,\mathbf{1}}^{-\frac{1}{2}} \left[D(w) + \Lambda(w)\right] H_{\lambda,\mathbf{1}}^{-\frac{1}{2}} g_\mathbf{1}(w)$$
$$\leq \left(\frac{\sigma_{\max}}{\lambda} + \frac{\sigma_{\max}(\sigma_{\max}+\lambda)}{2\lambda^2}\right) g_\mathbf{1}(w)^\top H_{\lambda,\mathbf{1}}^{-1} g_\mathbf{1}(w)$$
$$= \left(\frac{3\sigma_{\max}}{2\lambda} + \frac{\sigma_{\max}^2}{2\lambda^2}\right) \mathcal{I}_f(w),$$

which gives us

$$\mathcal{I}_f(w) + \mathrm{Err}_{f,3}(w) \leq \mathcal{I}_f^{\mathrm{Nt}}(w) \leq \left(1 + \frac{3\sigma_{\max}}{2\lambda} + \frac{\sigma_{\max}^2}{2\lambda^2}\right) \mathcal{I}_f(w) + \mathrm{Err}_{f,3}(w).$$

Note that $\mathrm{Err}_{f,2}(w) \stackrel{\mathrm{def}}{=} \frac{\sigma_{\max}^2}{2\lambda^2} \mathcal{I}_f(w)$ can be bounded as

$$|\mathrm{Err}_{f,2}(w)| = \left| \frac{\sigma_{\max}^2}{2\lambda^2} \mathcal{I}_f(w) \right|$$
$$\leq \frac{\sigma_{\max}^2}{2\lambda^2} \cdot |\mathcal{I}_f(w)|$$
$$\leq \frac{\sigma_{\max}^2}{2\lambda^2} \cdot \left| g_\mathbf{1}(w)^\top H_{\lambda,\mathbf{1}}^{-1} g_\mathbf{1}(w) \right|$$
$$\leq \frac{\|w\|_1^2 C_\ell^2 \sigma_{\max}^2}{2(\sigma_{\min} + \lambda)\lambda^2},$$

and $\text{Err}_{\text{Nt-act}}(w) = \mathcal{I}_f^*(w) - \mathcal{I}_f^{\text{Nt}}(w)$ grows as $O(1/\lambda^3)$ from Proposition 1, so we can also write

$$\mathcal{I}_f(w) + O\Big(\frac{1}{\lambda^3}\Big) \leq \mathcal{I}_f^*(w) \leq \Big(1 + \frac{3\sigma_{\max}}{2\lambda}\Big)\mathcal{I}_f(w) + O\Big(\frac{1}{\lambda^3}\Big).$$

$\square$

### E.6 The influence on a test point

**Corollary 1** (Restated). *Suppose* $f(\theta) = \theta^\top x_{\text{test}}$, *and define* $v_{\text{test}} \stackrel{\text{def}}{=} H_{\lambda,\mathbf{1}}^{-\frac{1}{2}} x_{\text{test}}$ *and* $v_w \stackrel{\text{def}}{=}$ $H_{\lambda,\mathbf{1}}^{-\frac{1}{2}} g_1(w)$. *Then under Assumptions 1 and 2,* $\mathcal{I}_f^{\text{Nt}}(w) = \mathcal{I}_f(w) + v_{\text{test}}^\top D(w) v_w$, *where* $D(w) = \big(I - H_{\lambda,\mathbf{1}}^{-\frac{1}{2}} H_\mathbf{1}(w) H_{\lambda,\mathbf{1}}^{-\frac{1}{2}}\big)^{-1} - I$ *is the* error matrix *from Proposition 2.*

*Proof.* Since $f(\theta) = \theta^\top x_{\text{test}}$ is linear, we have that for any $\theta \in \Theta$,

$$\nabla_\theta f(\theta) = x_{\text{test}},$$
$$\nabla_\theta^2 f(\theta) = 0,$$
$$C_{f,3} = 0.$$

This in turn implies that $\text{Err}_{f,3}(w) = 0$. Substituting these expressions into Proposition 2 gives us the desired result. $\square$

**Proposition 4** (Restated). *Consider a binary classification setting with* $y \in \{-1, +1\}$ *and a margin-based model with loss* $\ell(x, y; \theta) = \phi(y\theta^\top x)$ *for some* $\phi : \mathbb{R} \to \mathbb{R}_+$. *Suppose* $f(\theta) = \theta^\top x_{\text{test}}$ *and that the subset* $w$ *comprises* $\|w\|_1$ *identical copies of the training point* $(x_w, y_w)$. *Then under Assumptions 1 and 2, the Newton approximation* $\mathcal{I}_f^{\text{Nt}}(w)$ *is related to the influence* $\mathcal{I}_f(w)$ *according to*

$$\mathcal{I}_f^{\text{Nt}}(w) = \frac{\mathcal{I}_f(w)}{1 - \|w\|_1 \cdot \phi''(y_w \hat{\theta}(\mathbf{1})^\top x_w) \cdot x_w^\top H_{\lambda,\mathbf{1}}^{-1} x_w}.$$

*This implies the Newton approximation* $\mathcal{I}_f^{\text{Nt}}(w)$ *is bounded between* $\mathcal{I}_f(w)$ *and* $\big(1 + \frac{\sigma_{\max}}{\lambda}\big)\mathcal{I}_f(w)$.

*Proof.* From Corollary 1,

$$\mathcal{I}_f^{\text{Nt}}(w) = \mathcal{I}_f(w) + x_{\text{test}}^\top H_{\lambda,\mathbf{1}}^{-\frac{1}{2}} D(w) H_{\lambda,\mathbf{1}}^{-\frac{1}{2}} g_1(w).$$

With the additional assumptions on $w$ and $\ell(x, y; \theta)$, we have that

$$\nabla_\theta \ell(x, y; \theta) = y\phi'(y\theta^\top x)x,$$

$$g_1(w) = \sum_{i=1}^n w_i \nabla_\theta \ell(x_i, y_i; \hat{\theta}(\mathbf{1}))$$

$$= \sum_{i=1}^n w_i y_i \phi'(y_i \hat{\theta}(\mathbf{1})^\top x_i) x_i$$

$$= \sum_{i=1}^n w_i y_i \phi'_i x_i$$

$$= \|w\|_1 \, y_w \phi'_k x_w,$$

where in the last equality we use the assumption that we are removing $\|w\|_1$ copies of the point $(x_w, y_w)$. Similarly,

$$\nabla_\theta^2 \ell(x, y; \theta) = \phi''(y\theta^\top x)xx^\top,$$

$$H_\mathbf{1}(w) = \sum_{i=1}^n w_i \nabla_\theta^2 \ell(x_i, y_i; \hat{\theta}(\mathbf{1}))$$

$$= \sum_{i=1}^n w_i \phi''(y_i \hat{\theta}(\mathbf{1})^\top x_i) x_i x_i^\top$$

$$= \|w\|_1 \, \phi''_k x_w x_w^T.$$

We thus have

$$D(w) = \left(I - H_{\lambda,\mathbf{1}}^{-\frac{1}{2}} H_{\mathbf{1}}(w) H_{\lambda,\mathbf{1}}^{-\frac{1}{2}}\right)^{-1} - I$$

$$= \left(I - H_{\lambda,\mathbf{1}}^{-\frac{1}{2}} \|w\|_1 \phi_k'' x_w x_w^T H_{\lambda,\mathbf{1}}^{-\frac{1}{2}}\right)^{-1} - I$$

$$= \frac{H_{\lambda,\mathbf{1}}^{-\frac{1}{2}} \|w\|_1 \phi_k'' x_w x_w^T H_{\lambda,\mathbf{1}}^{-\frac{1}{2}}}{1 - \|w\|_1 \phi_k'' x_w^T H_{\lambda,\mathbf{1}}^{-1} x_w}$$

$$= \frac{\|w\|_1 \phi_k'' H_{\lambda,\mathbf{1}}^{-\frac{1}{2}} x_w x_w^T H_{\lambda,\mathbf{1}}^{-\frac{1}{2}}}{1 - \|w\|_1 \phi_k'' x_w^T H_{\lambda,\mathbf{1}}^{-1} x_w},$$

where the third equality comes from the Sherman-Morrison formula. Substituting $D(w)$ into Corollary 1, we obtain

$$\mathcal{I}_f^{\mathrm{Nt}}(w) = \mathcal{I}_f(w) + x_{\mathrm{test}}^\top H_{\lambda,\mathbf{1}}^{-\frac{1}{2}} D(w) H_{\lambda,\mathbf{1}}^{-\frac{1}{2}} g_{\mathbf{1}}(w)$$

$$= \mathcal{I}_f(w) + \frac{\|w\|_1 \phi_k'' x_{\mathrm{test}}^\top H_{\lambda,\mathbf{1}}^{-1} x_w x_w^T H_{\lambda,\mathbf{1}}^{-1} g_{\mathbf{1}}(w)}{1 - \|w\|_1 \phi_k'' x_w^T H_{\lambda,\mathbf{1}}^{-1} x_w}$$

$$= \mathcal{I}_f(w) + \frac{x_{\mathrm{test}}^\top H_{\lambda,\mathbf{1}}^{-1} \|w\|_1 y_w \phi_k' x_w \cdot \|w\|_1 \phi_k'' x_w^T H_{\lambda,\mathbf{1}}^{-1} x_w}{1 - \|w\|_1 \phi_k'' x_w^T H_{\lambda,\mathbf{1}}^{-1} x_w}$$

$$= \mathcal{I}_f(w) + \frac{x_{\mathrm{test}}^\top H_{\lambda,\mathbf{1}}^{-1} g_{\mathbf{1}}(w) \cdot \|w\|_1 z_k^T H_{\lambda,\mathbf{1}}^{-1} z_k}{1 - \|w\|_1 \phi_k'' x_w^T H_{\lambda,\mathbf{1}}^{-1} x_w}$$

$$= \mathcal{I}_f(w) + \frac{\mathcal{I}_f(w) \cdot \|w\|_1 z_k^T H_{\lambda,\mathbf{1}}^{-1} z_k}{1 - \|w\|_1 \phi_k'' x_w^T H_{\lambda,\mathbf{1}}^{-1} x_w}$$

$$= \frac{\mathcal{I}_f(w)}{1 - \|w\|_1 \phi_k'' x_w^T H_{\lambda,\mathbf{1}}^{-1} x_w}.$$

To bound the denominator, we first use the trace trick to rearrange terms

$$\|w\|_1 \phi_k'' x_w^T H_{\lambda,\mathbf{1}}^{-1} x_w = \mathrm{tr}\left(\|w\|_1 \phi_k'' x_w^T H_{\lambda,\mathbf{1}}^{-1} x_w\right)$$

$$= \mathrm{tr}\left(H_{\lambda,\mathbf{1}}^{-\frac{1}{2}} \|w\|_1 \phi_k'' x_w x_w^T H_{\lambda,\mathbf{1}}^{-\frac{1}{2}}\right)$$

$$= \mathrm{tr}\left(H_{\lambda,\mathbf{1}}^{-\frac{1}{2}} H_{\mathbf{1}}(w) H_{\lambda,\mathbf{1}}^{-\frac{1}{2}}\right).$$

Since $H_{\lambda,\mathbf{1}}^{-\frac{1}{2}} H_{\mathbf{1}}(w) H_{\lambda,\mathbf{1}}^{-\frac{1}{2}}$ has rank one under our assumptions, it only has at most one non-zero eigenvalue. We can therefore apply Lemma 1 to conclude that

$$\|w\|_1 \phi_k'' x_w^T H_{\lambda,\mathbf{1}}^{-1} x_w = \mathrm{tr}\left(H_{\lambda,\mathbf{1}}^{-\frac{1}{2}} H_{\mathbf{1}}(w) H_{\lambda,\mathbf{1}}^{-\frac{1}{2}}\right)$$

$$\leq \frac{\sigma_{\max}}{\sigma_{\max} + \lambda},$$

which in turn implies that $1 - \|w\|_1 \phi_k'' x_w^T H_{\lambda,\mathbf{1}}^{-1} x_w \geq \frac{\lambda}{\sigma_{\max}+\lambda}$, so

$$\frac{1}{1 - \|w\|_1 \phi_k'' x_w^T H_{\lambda,\mathbf{1}}^{-1} x_w} \leq \frac{\sigma_{\max} + \lambda}{\lambda} = 1 + \frac{\sigma_{\max}}{\lambda}.$$

$$\square$$

## Footnotes

[5]This experiment required us to retrain the model for every value of $\lambda$ and for every subset. Thus, for computational purposes, we omitted CDR and MultiNLI, and we selected a random 10% subset of MNIST's training set to use in place of all of MNIST.