[Reviews · NeurIPS 2019]

Reviewer 1



Originality - The work primarily builds on Koh and Liang's Influence functions work and extends it to evaluating influence over non-random groups of examples instead of single examples. The work is an interesting analytical extension of influence functions work. Contributions and related work are clearly specified and justified. Quality - The theoretical analyses appears sound although I did not go through the proofs in exact detail. The analyses shows empirical as well as theoretically the gaps in estimating influence using Koh and Liang's methods when used to analyze group effects. The analyses is mostly carried out in the regime where a large fraction of samples are removed. The analyses uses newton approximation of influence functions to bound a decomposition of the actual and estimated influence. Clarity - The paper is well written, comprehensive and empirical evaluation is adequate. Significance - The work in and of itself is interesting but significance to the field is low. Nonetheless the analyses has important implications of explainability and other related fields.

Reviewer 2



This paper examines the question of whether influence functions for determining the impact of a single training point on a model’s predictions generalize in the case where a group of points (e.g., a demographic group) that likely have cross-correlated effects are all removed. This is a fascinating and important problem. The authors demonstrate empirically that across five datasets the actual effect and the influence as approximated by the influence functions are correlated. This is an important result and validation for the real-world use of influence functions in important fairness domains. They then provide an analysis attempting to characterize the conditions under which the approximated influence and the actual effects are correlated, including nice experiments demonstrating that the Newton approximation is highly correlated to the actual effect.

Reviewer 3



# Originality As far as I can tell, the proposed use of influence functions is new. The experimental design, theoretical analysis, and case studies also appear to be novel. # Quality I think this paper is very well done. The empirical results are thorough, and the limitations of the theoretical analysis are clearly discussed. # Clarity The paper is well written; I enjoyed reading it. # Significance Several recent papers in the ML community have focused on influence functions. The paper presents another use for influence functions, and the case studies at the end of the paper make a compelling case for why group effect estimates are valuable. The paper also presents interesting new empirical and theoretical analyses that will likely be useful in other papers that study influence functions.

[Author Response · NeurIPS 2019]

We thank all of the reviewers for their thoughtful feedback, and will incorporate their suggestions into the next version
of our paper. We detail our responses to their comments below.

**R1.**   We thank R1 for their comments and will emphasize the broader implications of our work on model explainability.

**R2.**   R2 asked to contrast using (i) influence functions to measure the importance of training points with (ii) existing
techniques for measuring feature importance, namely Datta, Sen, & Zick, 2016; Adler et al., 2016; and Adebayo &
Kagal, 2016. These papers address a different problem setting from ours and their methods are correspondingly distinct.

The main difference is that the papers above seek to explain a *fixed* model $\theta$, whereas we examine how the *learned*
model $\hat{\theta}$ changes as a function of its training data. Our central issue is therefore reasoning about retraining the model,
which is not a concern shared by the papers above. Concretely, those papers consider the question: given a fixed model
$\theta$, how do its predictions on a test set $\mathcal{D}_{\text{test}}$ depend on the values of some feature $x_k$ in $\mathcal{D}_{\text{test}}$? They investigate this by
perturbing the value of $x_k$ in various ways (e.g., by randomizing $x_k$ for each test point $x$ in $\mathcal{D}_{\text{test}}$). In contrast, we are
given the training set $\mathcal{D}_{\text{train}}$ and our goal is determining the effect of removing groups of points in $\mathcal{D}_{\text{train}}$ on the learned $\hat{\theta}$.

Despite their differences, these methods could be complementary, as R2 suggested. For instance, if we find (with feature
importance methods) that a model depends heavily on some feature, we could use influence functions to identify the
training data most responsible for that dependence. We will include this discussion and we thank R2 for pointing it out.

**R3: Non-convex objectives.**   R3 asked if our empirical findings hold for non-convex models. Our initial experiments
are promising and suggest that this can be true; we will discuss this question in our next revision and plan to conduct a
more extensive study. We are grateful to R3 for highlighting this question. To properly respond, let us first provide
context on why influence functions and actual effects are classically only defined for convex models.

Recall that the actual effect $\mathcal{I}_f^*(w)$ of a subset $w$ measures the difference between (i) the original model $\hat{\theta}(\mathbf{1})$, which
minimizes the loss on the training data $\mathcal{D}_{\text{train}}$, and (ii) the new model $\hat{\theta}(\mathbf{1} - w)$, which minimizes the loss on $\mathcal{D}_{\text{train}}$ with
$w$ removed. Thus, for $\mathcal{I}_f^*(w)$ to be well-defined, there must be a unique model $\hat{\theta}(\mathbf{1})$ that globally minimizes the training
loss on $\mathcal{D}_{\text{train}}$, and likewise for $\hat{\theta}(\mathbf{1} - w)$. This condition is satisfied when the model is strongly convex. Similarly,
the influence $\mathcal{I}_f(w)$ is only well-defined when $\hat{\theta}(\mathbf{1})$ is unique and the model is strongly convex around it. Finally, to
measure the actual effect and influence, the models $\hat{\theta}(\mathbf{1})$ and $\hat{\theta}(\mathbf{1} - w)$ must not only be well-defined but computable.

Non-convex models unfortunately violate all of these requirements: the global minimizer $\hat{\theta}(\mathbf{1})$ may not be unique for a
given $\mathcal{D}_{\text{train}}$, and even if it were, we may not find it. For instance, neural networks are typically trained with SGD-based
methods that only guarantee convergence to a local minimum, so in general we cannot compute $\hat{\theta}(\mathbf{1})$ nor $\hat{\theta}(\mathbf{1} - w)$.

To address these issues, we propose augmenting the classical definitions as follows. Let the actual effect $\mathcal{I}_f^*(w, \theta_0, r)$ of
a subset $w$ given an initial trained model $\theta_0$ and a random seed $r$ to be the change in the model after removing $w$ and
retraining the model by starting from $\theta_0$ and running SGD with randomness $r$. Specifying the initial model $\theta_0$ sidesteps
the issue of $\hat{\theta}(\mathbf{1})$ being non-unique or impossible to compute, while specifying $r$ resolves the issue of the retrained
model $\hat{\theta}(\mathbf{1} - w)$ being ill-defined. Similarly, we augment the predicted effect $\mathcal{I}_f(w, \theta_0)$ to be the influence of $w$ around
the initial model $\theta_0$. Our goal is then to see if $\mathcal{I}_f(w, \theta_0) \approx \mathcal{I}_f^*(w, \theta_0, r)$ for all $r$.

We tested this with a multi-layer perceptron (MLP) with 2 hidden layers (128 and 32
nodes) on 10% of the MNIST 10-class dataset, choosing $\theta_0$ as a model trained from
scratch with an arbitrary random seed. As in our submission, we generated 70 coherent
groups of training points. For each group, we tried 50 values of $r$ but found negligible
variation between the actual effects (max difference of $3 \times 10^{-4}$ between $r$'s). Figure
S1 shows that influence is highly correlated with actual effects (for an arbitrary $r$) on
the test point with highest loss (Spearman $\rho = 0.96$), even in this non-convex setting.

**R3: Case studies.**   R3 is right to point out the distinction between removing certain
labeling functions (LFs) and crowdworkers from the training data, which we do in
our work, and actually changing the LFs or the crowdworker recruiting policy. For
example, it is possible that explicitly encouraging LF programmers to create LFs with
higher coverage could result in spurious LFs that lower model performance, even
though the naturally-obtained high-coverage LFs are the most helpful for the model.
Determining the actual effect of manipulating data collection will require systematic
user experiments, and we will clarify and emphasize this distinction. We thank R3 for
bringing this point up.

Figure S1: Predicted vs. actual effects on an MLP. Colors are as in our submission.

[Meta-Review · NeurIPS 2019]

As pointed out by the reviewers, these are the strengths and weaknesses of the paper: STRENGTHS The reviewers agree that the paper makes a contribution towards understanding the theoretical and empirical limitations of influence functions when samples are not missing at random. The empirical results show that influence functions are correlated with true influence even when the estimate error is high but they can also underestimate the true influence. The paper is well written and structured. FOR IMPROVEMENT The main reviewers’ concerns that need to be addressed in the revision are the ones discussed in the authors’ response: discussion on applicability to non-convex functions and different labeling functions, and contrasting with methods that specify groups as features.